# GOGGLE: GENERATIVE MODELLING FOR TABULAR DATA BY LEARNING RELATIONAL STRUCTURE

**Tennison Liu**
University of Cambridge
tl522@cam.ac.uk

**Zhaozhi Qian**
University of Cambridge
zq224@cam.ac.uk

**Jeroen Berrevoets**
University of Cambridge
jb2384@cam.ac.uk

**Mihaela van der Schaar**
University of Cambridge
Alan Turing Institute
mv472@cam.ac.uk

## ABSTRACT

Deep generative models learn highly complex and non-linear representations to generate realistic synthetic data. While they have achieved notable success in computer vision and natural language processing, similar advances have been less demonstrable in the tabular domain. This is partially because generative modelling of tabular data entails a particular set of challenges, including heterogeneous relationships, limited number of samples, and difficulties in incorporating prior knowledge. Additionally, unlike their counterparts in image and sequence domain, deep generative models for tabular data almost exclusively employ fully-connected layers, which encode weak inductive biases about relationships between inputs. Real-world data generating processes can often be represented using relational structures, which encode sparse, heterogeneous relationships between variables. In this work, we learn and exploit relational structure underlying tabular data (where typical dimensionality $d < 100$) to better model variable dependence, and as a natural means to introduce regularization on relationships and include prior knowledge. Specifically, we introduce GOGGLE, an end-to-end *message passing* scheme that jointly learns the relational structure and corresponding functional relationships as the basis of generating synthetic samples. Using real-world datasets, we provide empirical evidence that the proposed method is effective in generating realistic synthetic data and exploiting domain knowledge for downstream tasks.

## 1 INTRODUCTION

Learning generative models for synthetic data is an important area in machine learning with many applications. For example, synthetic data can be used to simulate settings where real data is scarce or unavailable [7, 10], support better supervised learning by increasing quality in datasets [8], improving robustness and predictive performance [69, 60], and promoting fairness [72]. Additionally, synthetic data is increasingly being used to overcome usage restrictions while preserving privacy [32, 78, 54].

*Deep generative models* have achieved notable success in approximating complicated, high-dimensional distributions as encountered in computer vision, natural language processing, and more [46]. A key contributor to this success is that learning architectures can easily exploit *relational inductive bias* that enhance the learning of joint distributions. Informally, relational inductive biases encode assumptions about the relational structure, which describes variables, and their relationships [3]. For example, image variables (pixels) have high covariance within a local region and relational rules that are invariant across regions—properties which are exploited by kernels in convolutional neural networks (CNN) to better model image distributions [42]. Similarly, sequence variables are highly dependent on sequentiality and relational rules are invariant across time-steps—recurrence relations leveraged by recurrent neural networks (RNNs) to capture distributions over time [27].

In this work, we hope to exploit similar relational inductive biases to better model real-world tabular datasets (where typical dimensionality $d < 100$). However, while images and sequences are

homogeneous data formats with known relational structure, tabular data commonly contain more heterogeneous relationships (e.g. variables are only correlated with a small subset of other variables), where the exact relational structure is obscured by domain-specific knowledge [62]. Without an obvious relational structure, deep generative models almost exclusively employ multilayer perceptrons (MLP) to learn representations on tabular data. This is less than ideal as MLPs encode virtually no relational information—indeed, all variables can interact to determine any other variable's value.

However, this all-to-all relational structure is often unnecessary, the *data generating process* (DGP) of tabular data is better described using sparse relational structures [39, 4]. Variable dependencies can be more accurately captured by considering them as edges and learning representations over the resulting relational structure. We hypothesize that generative models exploiting the relational structure can more adequately address certain challenges that arise during modelling distributions for tabular data, including ▶ **heterogeneous relationships** between variables, ▶ **smaller datasets**, which are more prone to overfitting, and ▶ the lack of mechanism to incorporate **prior knowledge** that can improve modelling performance.

**Contributions.** We introduce Generative MOdellinG with Graph LEarning (GOGGLE), an end-to-end framework that learns an approximate relational structure as the foundation of generative modelling. More specifically, we devise a general *message passing* scheme that models tabular data by jointly learning (1) the *relational structure* and (2) the corresponding *functional relationships* (*dependencies*) in the learned structure. To the best of our knowledge, this is the first work to jointly learn the relational structure and the parameters of a generative model to model tabular data. Additionally, we propose regularization on variable dependencies to reduce model overfitting on smaller tabular datasets and propose a simple mechanism to include prior knowledge into the generative process.

We demonstrate the advantages of our approach in a series of experiments on multiple real-world datasets. We employ both qualitative and quantitative approaches to demonstrate that GOGGLE achieves consistent improvements over state-of-the-art benchmarks in generating synthetic data and exploiting prior knowledge for better downstream performance.

## 2 CHALLENGES IN TABULAR DATA GENERATION

While deep generative models have seen notable success in image and sequence domains, tabular data is ubiquitous in many salient applications, including medicine, finance, and economics. Generative modelling for tabular data presents a distinct set of challenges, which are largely open research questions. Here, we highlight them in turn:

1. **Complicated relational structure.** Tabular data commonly contain heterogeneous relational structures, including sparse dependencies (variables only dependent on a small subset of other variables), and heterogeneous functional relationships (dependencies) between variables [62]. Unlike in images and sequences, where the relational structures (locality and sequentiality respectively) are better understood (and arguably generalizable), variable dependencies in tabular datasets are domain specific and rarely known.

2. **Overfitting and memorization.** Modern deep generative models are over-parameterized, thus requiring large datasets to learn the underlying distribution without overfitting [1]. This is especially demanding on tabular datasets that are smaller and where collection is difficult/expensive. Real-world DGPs are sparse in structure, and we address overfitting concerns by enforcing sparsity in variable dependencies, thus achieving a *regularization* effect by restricting the hypothesis space. Additionally, as variables can only be generated using their neighborhoods, the model is incentivized to find informative neighbors.

3. **Domain knowledge.** In many fields, such as medicine or social sciences, we have rich domain knowledge on dependence between variables, sparsity, or importance of specific variables (i.e. degree of connectivity [74]). Incorporating prior knowledge is especially useful in practical settings where we may not have large datasets but can obtain expert knowledge to aid in model learning. As far as we know, this is a capability that is currently lacking in tabular deep generative models. Our generative process takes into account the relational structure, allowing a diverse range of (partial) *domain knowledge* to be incorporated.

**A distinction.** We emphasize that the goal of our work is not *probabilistic structure discovery*, which aims to discover the *unique* probabilistic graph from observed data [15, 85]. As there is

Table 1: **Overview of generative models in the tabular domain.** Comparisons are made on underlying **model class** and deep learning **module**, generative **model distribution** $p_\theta$, and the following desiderata: model is capable of **(1)** generating in-distribution samples, **(2)** regularizing variable dependencies, **(3)** incorporating prior knowledge.

| Model | Model Class/ Module | Generative model $p_\theta$ | (1) | (2) | (3) |
|---|---|---|---|---|---|
| *Non-neural methods* | | | | | |
| BN | BN / None | $\prod_i^d p_\theta(x_i \| Pa(x_i))$ | ✓ | ✓ | ✓ |
| MM | MM / None | $\sum_{k=1}^{K} \pi_k \mathcal{N}(\mu_k, \sigma_k; \theta)$ | ✓ | ✗ | ✗ |
| *Neural methods* | | | | | |
| CopulaGAN | GAN / MLP | | ✓ | ✗ | ✗ |
| TableGAN | GAN / MLP | $\int p_\theta(x\|z)p(z)\,dz \; ; \; p(z) \in \mathcal{P}_Z$ | ✓ | ✗ | ✗ |
| CTGAN | GAN / MLP | | ✓ | ✗ | ✗ |
| TVAE | VAE / MLP | | ✓ | ✗ | ✗ |
| GOGGLE | VAE / MPNN | $\prod_i^d p_\theta(x_i\|\mathcal{N}(i); G) \; ; \; G \in \mathcal{G}$ | ✓ | ✓ | ✓ |

generally a set of plausible graphs that have equal likelihood on the training data, such methods make assumptions about the graph type, variable distribution, and functional relations, which are necessary to disambiguate a unique graph. In contrast, the aim of GOGGLE is to learn an approximate relational structure to guide generative modelling (see Appendix D).

## 3 RELATED WORKS

Traditionally, generative modelling was performed through explicit parameterisation and estimation of a joint probability distributions. Examples of this include Bayesian networks (BN) [82, 33], mixture models (MM) [56, 29] and copulas (CP) [68, 49]. These models are limited in more complex distributions, where it is difficult to learn high-dimensional dependencies correctly.

Recent advances in deep generative modelling (including VAEs [35], GANs [23], normalizing flows (NFs) [55] have seen models that can generate realistic synthetic data from complex, high-dimensional distributions. Specifically for tabular synthetic data, [76] introduced CTGAN and TVAE, which are GAN-based and VAE-based models respectively that tackled many practical issues of modelling tabular data, including multimodality and mixed data types. Similarly, TableGAN [47] employed a GAN framework with an auxiliary classifier to predict the label of a generated sample. medGAN [10] and ehrGAN [7] are more specialist methods developed for healthcare specifically. Instead of generating synthetic data, neural methods have also been employed to perform data imputation [22, 80]. Perhaps most similar to our work, [37] (and its tabular variant [75]) learn a causal graph in the representation space to generate synthetic data, but assume access to the true causal graph.

The aforementioned methods resort to MLPs to model complex dependencies in an all-to-all fashion. Additionally, they employ regularization in the weight space (e.g. L2 regularization [58] and Dropout [66]) to reduce overfitting. These effects, as we will discover in §5, are unlikely to be optimal for generative performance. Our work is also related to the field of *relational inference*, which seeks to infer relationships between objects from observation data alone. Representative works include [36], which seeks to infer interactions between objects in interacting systems, and [24, 25] that infers relationships from omic interactions. In these works, a correctly recovered relational structure *is* the object of inference. This stands in stark contrast to our work, where a partially correct structure is satisfactory for our purposes. Indeed, as we shall show later, even learning a partially correct structure can greatly improve synthetic data performance (see Appendix D for further discussion). The key contribution of this work is in learning the relational structure jointly with deep representation learning modules to improve generative modelling. We provide an overview in Table 1, evaluating related methods based on their ability to address the previously described challenges.

## 4 GOGGLE: GENERATIVE MODELLING WITH GRAPH LEARNING

### 4.1 PROBLEM FORMULATION

**Generative modelling.** Generative modelling for tabular data attempts to uncover a probability distribution $p_X$ over $X \in \mathcal{X} \subseteq \mathbb{R}^d$.[1] We have access to a training dataset $\mathcal{D}$, which consists of $N$ i.i.d. samples $x \sim p_X$. The goal is to learn the parameters $\theta$ of a generative model such that the

---

[1] Generative modelling for high-dimensional datasets, as those encountered in genomics (where $d$ typically$> 1000$) is outside the scope of this work.

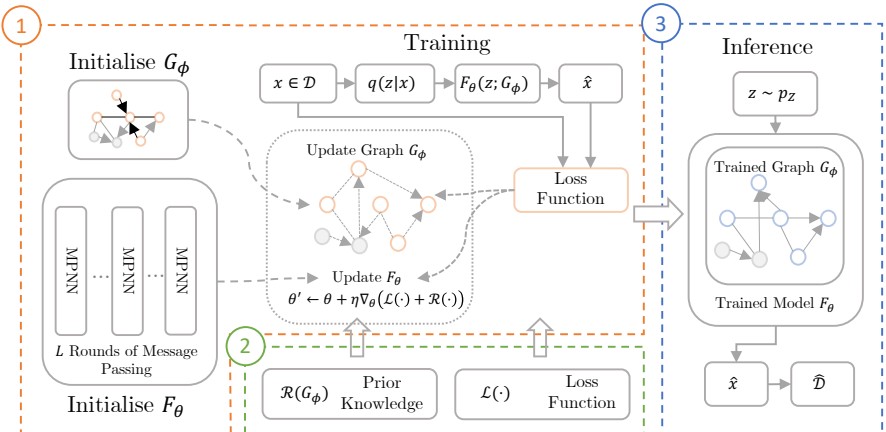

Figure 1: **Key components of GOGGLE Framework.** ① Simultaneous learning of relational structure $G_\phi$ and $F_\theta$ s.t. generative process respects relational structure. ② Injection of prior knowledge and regularization on variable dependence. ③ Synthetic sample generated using $\hat{x} = F_\theta(z; G_\phi)$, $z \sim p_Z$.

model distribution $p_\theta$ is close to $p_X$. Most deep generative models employ a set of noise variables $Z \in \mathcal{Z} \subseteq \mathbb{R}^q$, which follows a *tractable* distribution $p_Z \in \mathcal{P}_Z$ (e.g. Gaussian distribution). Instead of learning $p_X$ directly, the model learns a mapping $g_\theta : \mathcal{Z} \to \mathcal{X}$ where the transformed variable $g_\theta(z)$ has the same distribution as $X$. The mapping $g_\theta$ can either be a surjective function (as in GANs [23]), a bijective function (as in normalizing flows [55]), or a stochastic mapping (as in VAEs [35]) [45].

**Relational structures as graphs.** A graph $G = (X, E)$ is a representation of the relational structure, where $X$ denotes random variables and $E$ edges that encode relationships between variables. The most general type of graph is a *mixed graphs*, which contain both directed and undirected edges, where an undirected edge e.g. $X_i - X_j$ is encoded as two directed edges $X_i \to X_j$ and $X_j \to X_i$. Importantly, a graph admits a *sparse* and *compact* representation, which underlies many real-world generative processes [6, 41, 20]. Specifically, we can say that a variable $X_i$ is only dependent on other variables with directed edges into $X_i$, $\mathcal{N}(i) = \{X_j : (X_j \to X_i) \in E\}$. In light of simplicity, we will refer to $\mathcal{N}(i)$ as the neighborhood of $X_i$. Formally, we can state that each variable is determined by a specific *functional relationship*: $X_i = f_i(\mathcal{N}(i), \varepsilon_i)$, where $\varepsilon_i$ is a noise variable.

## 4.2 OVERVIEW

We hypothesize that sparse dependencies in a tabular dataset can be more accurately captured through a relational structure. To that end, we propose a novel framework of generative modelling that is guided by an underlying relational structure. While tabular generative models are conventionally constructed with MLPs, we introduce a *message passing* scheme that operates on a learned graph, better capturing heterogeneous variable relationships, and allowing regularization and prior knowledge to be injected through the adjacency matrix. As the underlying structure is not known a priori (or partially observed at best) in realistic settings, we design an end-to-end generative model to learn the relational structure simultaneously with the corresponding functional relationships.

Figure 1 provides a schematic overview of GOGGLE. The four key components are: ▶ **learnable relational structure**, which is parameterized by a graph and represented using a *weighted* adjacency matrix $G_\phi$ that indicates dependence between variables; ▶ **generative model** $F_\theta(\cdot)$ that maps from noise vector $z \sim p_Z$ such that the transformed variable $F_\theta(z; G_\phi)$ has the same distribution as $X$. The way the relational structure and the generative model interacts is that the generative model only allows the set of relations deemed important by the learned relational structure $G_\phi$ to influence the generation of a variable. Additionally, there are the ▶ **loss function** $\mathcal{L}(\cdot)$ and **regularization term** $\mathcal{R}(G_\phi)$ used to train the model and encode prior knowledge, respectively.

## 4.3 LEARNING THE RELATIONAL STRUCTURE

Learning the relational structure entails exploring possible adjacency matrices based on relationships between variables. In the absence of informative prior knowledge, we make the key (but minimal) assumption of *sparsity* (viz. Assumption 1), or that the relational structure is sparser than a fully-

connected graph. Intuitively, one can consider MLPs that represent all-to-all relationships as being described by a *fully-connected* relational structure. The sparsity assumption encourages learning of a sparse structure that also determines the most informative neighborhood that generates each variable with the highest likelihood. When we have some prior knowledge about variable dependencies, we will incorporate it through the adjacency matrix and a regularization term in the loss function.

**Assumption 1** (Graph sparsity). *We assume there exists a graph $G$ such that the adjacency matrix $A = \mathcal{A}(G) \in \{0,1\}^{d \times d}$ has at least one instance where $A_{ij} = A_{ji} = 0$, indicating variables $X_i$ and $X_j$ are not connected.*

In our graphs, nodes are random variables and edges denote dependence between them. We represent an undirected edge $X_i - X_j$ as two directed edges, one from $X_i \to X_j$, and one from $X_j \to X_i$. Specifically, we learn the entries in the unnormalized adjacency matrix, $\tilde{G}_\phi \in \mathcal{R}^{d \times d}$, which is then normalized through an element-wise `sigmoid`$(\cdot)$ function to obtain $G_\phi \in [0,1]^{d \times d}$. Intuitively, $G_\phi[i,j]$ represents the strength of the dependence between $X_i \to X_j$.[2]

## 4.4 LEARNING THE FUNCTIONAL RELATIONSHIPS

The generative process must be consistent with the relational dependencies specified by the learned relational structure $G_\phi$. We learn a generative model $F_\theta : \mathcal{Z} \to \mathcal{X}$ by learning the mapping from a tractable noise distribution (that can be easily sampled from) to the data distribution. For the ease of exposition, we can informally say that the generative model describes the set of functional relationships between variables $F_\theta = \{f_1, \cdots, f_d\}$, where each $f_i$ describes how $X_i$ depends on its neighborhood $\mathcal{N}(i)$ in $G_\phi$.

**Learning challenges.** Learning $F_\theta$ leads to two challenges. ▶ **Computational complexity:** $G_\phi$ is continuously updated during training, possibly resulting in different dependencies (different set of $d$ functional relationships) that has to be learned, limiting the scalability of the approach. ▶ **Cycles in the graph:** Consider an undirected edge between two variables $X_i - X_j$. In this example, $X_i$ is in the neighborhood of $X_j$, and the generation of $X_i$ will have to depend on the values of $X_j$. The reverse is also true. This highlights that $F_\theta(\cdot)$ should be able to work with potential cycles in the $G_\phi$ to ensure the functional relationships in the model are faithful to the learned graph.

**A flexible parameterization.** To address these challenges, we employ a *message passing neural network* (MPNN) as a flexible and expressive parameterisation of the functional relationships model [21]. This scheme addresses the aforementioned challenges by naturally handling cycles in the graph and relaxing the computational burden of learning functional relations by applying a common information propagation procedure. The proposed message passing scheme performs $L$ rounds of message passing and generates each variable $\hat{x}_i$ from an initial embedding $h_i^{(0)}$, which is constructed using the noise term $z_i$. Each round $l \in [L]$ of message passing is defined in terms of a *message* function $\sigma^{(l)}(\cdot)$, an *aggregation* function $\oplus^{(l)}(\cdot)$ and an *update* function $\gamma^{(l)}(\cdot)$.

During each round $l$, a message $m_j^{(l)}$ is constructed for each of the variables in the neighborhood $\forall j \in \mathcal{N}(i)$, using the message function $\sigma^{(l)}$:

$$m_j^{(l)} = \sigma^{(l)} \left( h_j^{(l-1)} \right) \tag{1}$$

Then, the aggregation function $\oplus^{(l)}$ combines all incoming messages from the neighborhood through weighing each individual message by the learned weight in $G_\phi$:

$$h_{\mathcal{N}(i)}^{(l)} = \oplus^{(l)} \left( \{ G_{j,i} m_j^{(l)}, \forall j \in \mathcal{N}(i) \} \right) \tag{2}$$

The variable embeddings $h_i^{(l)}$ are then updated using the aggregated neighborhood messages and its previous embedding $h_i^{(l-1)}$ through the update function $\gamma^{(l)}$:

$$h_i^{(l)} = \gamma^{(l)} \left( h_{\mathcal{N}(i)}^{(l)}, h_i^{(l-1)} \right) \tag{3}$$

Each round of message passing exploits the relationships and sparsity in the learned structure by generating each variable using solely information from variables it depends on.

---

[2] One could also interpret $G_\phi[i,j]$ as the probability of an edge existing from $X_i \to X_j$ and obtain a binary adjacency matrix by sampling, where the parameters could be learned through the Gumbel-Softmax trick [31].

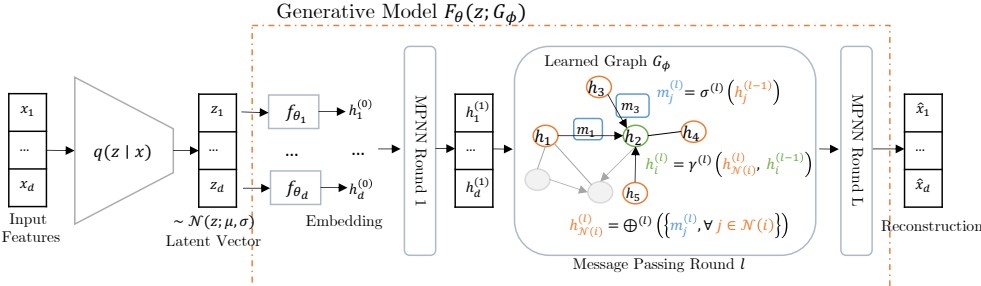

Figure 2: **Generative architecture.** The generative model is consistent with the learned relational structure by generating each variable using its neighborhood. GOGGLE adopts a VAE architecture with a MPNN to gradually generate synthetic data through $L$ rounds of message passing.

To obtain the initial node embeddings $h^{(0)}$, we add the variable index as a one-hot encoded vector to the latent variable, which is then transformed using an embedding function, i.e. $h_i^{(0)} = f_{\theta_i}(\texttt{concat}[z_i, \mathbb{1}_i])$. Here, $\mathbb{1}_i$ is the one-hot vector for the variable $i$, $\texttt{concat}[\cdot]$ is the concatenation function and $f_{\theta_i}(\cdot)$ is the embedding function implemented using a single-layer MLP. This transformation encodes meaningful variable-specific information in messages and embeddings. Using this scheme, the learning problem is drastically simplified from learning $d$ functional relationships in each round to just two message passing functions.

Lastly, $\sigma^{(l)}$ and $\gamma^{(l)}$ are arbitrary, learnable functions, and $\oplus^{(l)}$ is a permutation-invariant function, including mean, max pooling, or sum. In other words, $F_\theta = \{(\sigma^{(l)}, \gamma^{(l)}, \oplus^{(l)}) \forall l \in L\}$. The node embeddings after the last round are taken to be the generation, $\hat{x}_i = h_i^{(L)}$. The number of rounds $L$ of message passing, the choice of $\sigma, \gamma, \oplus$, and the construction of the initial embeddings $h^{(0)}$ are design choices that should be made to suit each task specifically. Additionally, the scheme can be *synchronous*, which is useful when all the variables are being generated jointly, or *asynchronous*, which is more suitable when the variables are being generated sequentially.

## 4.5 PUTTING IT TOGETHER

We adopt a VAE style architecture [35] and perform amortized inference on the initial noise vectors.[3] Specifically, the noise vector is sampled from an encoder using the reparameterization trick $z = \mu + \sigma\varepsilon, \varepsilon \sim \mathcal{N}(0, I)$. The functional relationships model $F_\theta(\cdot)$ plays the role of the decoder to generate $\hat{x} = F_\theta(z; G_\phi) \sim p(x|z)$. The exact design of the method is visually illustrated in Figure 2.

**Loss function and regularization.** The generative model and learnable graph are jointly trained by combining the classic ELBO loss function with a graph regularization term $\mathcal{R}(G_\theta)$. The training objective is described below, where $\lambda$ is a hyperparameter controlling the regularization strength.

$$\mathcal{J}(\theta, \phi) = ELBO(X, \hat{X}; \theta, \phi) + \lambda\mathcal{R}(G_\phi)$$
$$= \underbrace{\mathbb{E}_{q(z|x)}[\ln p_\theta(x|z; G_\phi)]}_{\text{Likelihood}} - \underbrace{D_{KL}[q(z|x)||p(z)]}_{\text{KL}} + \underbrace{\lambda\mathcal{R}(G_\theta)}_{\text{Regularisation}} \qquad (4)$$

A key advantage of GOGGLE is that we can easily incorporate prior knowledge about the graph through the regularization term $\mathcal{R}(G_\phi)$. This term will encourage the adjacency matrix to have certain characteristics. In general, a sparsity prior $\mathcal{R}(G_\phi) = ||G_\phi||_p$ will reward sparse graphs, and partial knowledge of feature relationships can be encoded through the prior $\mathcal{R}(G_\phi) = ||G_\phi - G_0||_p$, where $||\cdot||_p$ denotes the $p$-norm. We provide a more thorough discussion on the types of prior knowledge that can be incorporated in Appendix B.

**Training and generation.** We implement message construction and embedding update functions using single-layer ReLU-activated MLP. Specifically, $\sigma^{(l)} = ReLU\left(W_m^{(l)} \times h_j^{(l-1)}\right)$ and $\gamma^{(l)} = ReLU\left(W_u^{(l)} \times \texttt{concat}\left[h_{\mathcal{N}(i)}^{(l)}, h_i^{(l-1)}\right]\right)$. Here, $W_m$ and $W_u$ are weight matrices of their respective MLPs. Additionally, we aggregate messages using the mean, which can be viewed as taking a weighted average over messages from a neighborhood.

---

[3]We note that while we consider a VAE architecture, the appropriate generative architecture and loss functions (e.g. GANs, normalizing flows) should be designed with the application in mind and such extensions are left for future works.

We use a standard normal prior for $p_Z = \mathcal{N}(0, I)$. The adjacency matrix is initialized as a fully-connected graph, where edge weights are gradually refined during learning. In practice, we apply a hard threshold on the learned adjacency matrices, where entries $< 0.1$ are zeroed out (similar to ReLU activation). In the absence of prior knowledge, we impose a sparsity penalty $\mathcal{R}(G_\phi) = ||G_\phi||_1$. Once the training is complete, a synthetic sample can be generated by first sampling a noise vector $z \sim p_Z$ and passing it through the functional relationships model: $\hat{x} = F_\theta(z; G_\phi)$.

### 4.6 A Remark on Data Augmentation

Moreover, and this might be of independent interest, we show that by using the blueprint presented, we can also construct a generative model that can perform *data augmentation*. We consider data augmentation as the ability to conditionally generate synthetic data when conditioned on *any* variable to be a specific value, i.e. $\hat{x} = F_\theta(z, X_i = x_i; G_\phi)$. To do so, we assume the DGP can be represented by a *directed, acyclic graph* (DAG), meaning that variables are generated sequentially (*asynchronous message passing*), following a topological order $\pi_{G_\phi}$ obtained from the learned graph. Specifically, the generation of each variable is described through a structural equation model: $\hat{x}_i = f_i(\hat{\mathcal{N}}_i, z_i)$ [50]. During generation, we can condition specific variables on a particular value to allow sampling from the conditional distribution. We elaborate more on this method in Appendix A.

## 5 Experiments

The core claim in this work is that exploring relational inductive biases can better capture the sparsity and heterogeneous relationships in tabular data, and consequentially lead to enhanced learning of generative models. We quantitatively evaluate aspects of our method to support this claim:

1. **Synthetic data quality**: *How good is the synthetic data?* §5.1 quantitatively evaluates the characteristics of the synthetic data with respect to a variety of state-of-the-art benchmarks.
2. **Prior knowledge**: *Does prior knowledge improve performance?* §5.2 evaluates whether prior knowledge about variable dependence can improve generation performance.
3. **Gains**: *Why does it work?* §5.3 investigates the dynamics of relational structure learning and generative model, and to what extent the relational structure contributes to performance gains.

In the interest of limited space, we attach additional results in Appendix C. Specifically, we include: 4. **Additional datasets**: evaluating synthetic data performance on 6 more datasets; 5. **t-SNE and graph visualizations**: t-SNE visualization on synthetic datasets to qualitatively investigate quality [73]; we also examine the graphs learned in our experiments; 6. **Sensitivity analysis**: to better evaluate the performance of our method on different dataset sizes and number of features; 7. **Data augmentation**: assesses an alternative implementation of our method to perform data augmentation.

**Benchmarks.** We compare against state-of-the-art tabular synthetic data models, including Bayesian Networks (BN) [51] and GAN-based models: CTGAN [76] and TableGAN [47]; VAE-based models: TVAE [49]; and normalising flows NFLOW [55].

Following the experiment design in recent works [49, 76], we employ 8 real-world datasets from the UCI repository [16], and 2 datasets from the BN repository [39]. We employ datasets with different number of samples (ranging from 569 to 581012) and different feature counts (ranging from 12 to 168) to gain a better understanding of our method's performance profile. We provide additional information about benchmarks, datasets, hyperparameters, and evaluation methods in Appendix B. For all results, we report mean $\pm$ std averaged over 10 runs. Our code is provided on GitHub.[4]

### 5.1 Evaluation of Synthetic Data

To assess the quality of synthetic data, we observe three desiderata (similar to [79, 76]): ▶ **quality**—samples should be *realistic*, cover the data distribution, and be generalized. We evaluate using the three-dimensional metric ($\alpha$-precision, $\beta$-recall, authenticity) proposed in [1], reporting their average. ▶ **detection**—samples should be indistinguishable from the real data. We report the AUROC performance of three post-hoc classifiers to distinguish real and generated samples. Finally, ▶ **utility**—samples should be just as useful as the real data on predictive tasks (i.e. train-on-synthetic,

---

[4] https://github.com/tennisonliu/GOGGLE; https://github.com/vanderschaarlab/GOGGLE

Table 2: **Quality, detection, and utility of synthetic data.** Bold indicates the best performance.

| | Dataset | Adult | Breast | Covertype | Credit |
|---|---|---|---|---|---|
| **Quality** ($\uparrow$) (higher is better) | BN | $0.61 \pm 0.04$ | $0.59 \pm 0.02$ | $\mathbf{0.59 \pm 0.02}$ | $0.61 \pm 0.03$ |
| | MM | $0.58 \pm 0.02$ | $0.48 \pm 0.03$ | $0.52 \pm 0.06$ | $0.57 \pm 0.03$ |
| | CTGAN | $0.57 \pm 0.01$ | $0.58 \pm 0.02$ | $0.59 \pm 0.04$ | $\mathbf{0.61 \pm 0.07}$ |
| | TableGAN | $0.59 \pm 0.01$ | $0.35 \pm 0.06$ | $0.63 \pm 0.04$ | $0.61 \pm 0.06$ |
| | TVAE | $0.59 \pm 0.03$ | $0.48 \pm 0.02$ | $0.60 \pm 0.07$ | $0.53 \pm 0.03$ |
| | NFLOW | $0.61 \pm 0.02$ | $0.53 \pm 0.02$ | $0.58 \pm 0.02$ | $0.58 \pm 0.03$ |
| | GOGGLE | $\mathbf{0.62 \pm 0.02}$ | $\mathbf{0.61 \pm 0.01}$ | $0.59 \pm 0.06$ | $\mathbf{0.61 \pm 0.07}$ |
| **Detection** ($\downarrow$) (lower is better) | BN | $0.69 \pm 0.03$ | $0.69 \pm 0.02$ | $\mathbf{0.66 \pm 0.03}$ | $0.70 \pm 0.02$ |
| | MM | $0.72 \pm 0.04$ | $0.71 \pm 0.04$ | $0.80 \pm 0.05$ | $0.75 \pm 0.05$ |
| | CTGAN | $0.70 \pm 0.03$ | $0.70 \pm 0.02$ | $0.72 \pm 0.03$ | $0.71 \pm 0.04$ |
| | TableGAN | $0.76 \pm 0.03$ | $0.80 \pm 0.04$ | $0.79 \pm 0.05$ | $0.69 \pm 0.04$ |
| | TVAE | $0.73 \pm 0.04$ | $0.72 \pm 0.03$ | $0.72 \pm 0.05$ | $0.77 \pm 0.03$ |
| | NFLOW | $0.78 \pm 0.03$ | $0.72 \pm 0.05$ | $0.79 \pm 0.05$ | $0.74 \pm 0.03$ |
| | GOGGLE | $\mathbf{0.68 \pm 0.04}$ | $\mathbf{0.68 \pm 0.03}$ | $0.72 \pm 0.03$ | $\mathbf{0.69 \pm 0.03}$ |
| **Utility** ($\uparrow$) (higher is better) | BN | $\mathbf{-0.01 \pm 0.00}$ | $-0.08 \pm 0.01$ | $-0.06 \pm 0.00$ | $-0.06 \pm 0.00$ |
| | MM | $-0.06 \pm 0.01$ | $-0.08 \pm 0.01$ | $-0.12 \pm 0.03$ | $-0.10 \pm 0.02$ |
| | CTGAN | $-0.05 \pm 0.00$ | $-0.06 \pm 0.09$ | $-0.06 \pm 0.01$ | $-0.05 \pm 0.03$ |
| | TableGAN | $-0.04 \pm 0.01$ | $-0.31 \pm 0.08$ | $-0.18 \pm 0.04$ | $-0.21 \pm 0.05$ |
| | TVAE | $-0.08 \pm 0.01$ | $-0.12 \pm 0.02$ | $-0.06 \pm 0.01$ | $-0.06 \pm 0.00$ |
| | NFLOW | $-0.04 \pm 0.01$ | $-0.07 \pm 0.01$ | $-0.08 \pm 0.02$ | $-0.13 \pm 0.03$ |
| | GOGGLE | $\mathbf{-0.01 \pm 0.00}$ | $\mathbf{-0.04 \pm 0.01}$ | $\mathbf{-0.05 \pm 0.02}$ | $\mathbf{-0.04 \pm 0.01}$ |

test-on-real) [19]. Utility is evaluated based on average AUROC change of three downstream prediction models trained on synthetic data relative to real data.

As indicated in Table 2, GOGGLE consistently generates higher quality and lower detection synthetic data relative to benchmarks. This improvement is especially noticeable on smaller datasets (i.e. breast and credit). In this regime, conventional generative models will overfit to the limited $\mathcal{D}_{train}$, whereas GOGGLE exploits sparsity to achieve better generalization performance. We note that BN emerge as our strongest competitors in this setting (also noted in [76]),

Table 3: **Average rank of models.** GOGGLE consistently achieves superior synthetic data quality.

| | Quality ($\uparrow$) | Detection ($\downarrow$) | Utility ($\uparrow$) |
|---|---|---|---|
| CTGAN | 4.9 | 4.1 | 4.1 |
| TableGAN | 4.8 | 4.8 | 4.5 |
| TVAE | 3.6 | 4.1 | 3.1 |
| BN | 1.7 | 2.1 | 2.3 |
| NFLOW | 2.9 | 3.5 | 3.2 |
| GOGGLE | **1.1** | **1.4** | **1.3** |

frequently out-performing deep generative models. The logical hypothesis is that BN generalize well to smaller datasets due to their stricter assumptions on the underlying graph, adding a strong regularization effect. However, on larger-scale datasets, learning a high-quality BN is more difficult. GOGGLE, however, maintains superior performance on larger, more complex datasets as our assumptions are less restrictive. On the utility aspect, indicates superior predictive performance on all datasets. Remarkable, the predictive scores on GOGGLE's synthetic data are very close to those on real data. We evaluate on 6 additional datasets in Appendix C.1, and reported the average rank of each method across all 10 datasets in Table 3.

## 5.2 GENERATION WITH PRIOR KNOWLEDGE

Next, we investigate whether incorporating partial knowledge through the relational structure can lead to enhanced performance. Specifically, we regularize the learned graph $G$ with partial prior knowledge $G_0$, i.e. $\mathcal{R}_G = ||G - G_0||_1$. This form of incorporating prior knowledge reflects that we are not completely *confident* in our prior knowledge.

We evaluate on ECOLI an MAGIC, where we know the true underlying graph a-prior. Specifically, we investigate three settings: ▶ **zero knowledge** $GOGGLE_0$, where no domain knowledge is used, ▶ **partial knowledge** $GOGGLE_{50}$, where $50\%$ of edges are known (by randomly sub-sampling $50\%$ of the edges), and ▶ **complete knowledge** $GOGGLE_{100}$ where the complete graph is known. We are interested in evaluating in each setting when different amounts of data are available to train a model. We visually depict the effect of prior knowledge on synthetic data quality in Figure 3.

The DGPs for both datasets (known *a-prior*) are Gaussian Bayesian Networks, matching exactly the assumptions of the BN baseline, giving BN the upper hand as they have the correct model specification. GOGGLE, even with partial knowledge, achieves significant performance gain when no prior knowledge is employed. The performance gains from prior knowledge are especially noticeable in lower data regimes, i.e. when $|\mathcal{D}_{train}| < 500$ samples. This highlights a key advantage of our

Table 4: **Ablation study.** Bold indicates the best performance.

| | Adult | | | Credit | | |
|---|---|---|---|---|---|---|
| | **Quality** ($\uparrow$) | **Detection** ($\downarrow$) | **Utility** ($\uparrow$) | **Quality** ($\uparrow$) | **Detection** ($\downarrow$) | **Utility** ($\uparrow$) |
| GOGGLE-ER | $0.41 \pm 0.07$ | $0.81 \pm 0.05$ | $-0.54 \pm 0.08$ | $0.46 \pm 0.03$ | $0.71 \pm 0.03$ | $-0.05 \pm 0.01$ |
| GOGGLE-COV | $0.53 \pm 0.06$ | $0.74 \pm 0.05$ | $-0.07 \pm 0.01$ | $0.55 \pm 0.04$ | $0.67 \pm 0.07$ | $-0.03 \pm 0.01$ |
| GOGGLE-BN | $0.56 \pm 0.04$ | $0.69 \pm 0.05$ | $-0.05 \pm 0.01$ | $0.41 \pm 0.09$ | $0.67 \pm 0.04$ | $-0.06 \pm 0.01$ |
| GOGGLE-DENSE | $0.49 \pm 0.09$ | $0.69 \pm 0.04$ | $-0.14 \pm 0.03$ | $0.56 \pm 0.03$ | $0.62 \pm 0.08$ | $-0.07 \pm 0.01$ |
| GOGGLE-NMP | $0.53 \pm 0.07$ | $0.72 \pm 0.08$ | $-0.15 \pm 0.03$ | $0.60 \pm 0.04$ | $0.65 \pm 0.05$ | $-0.03 \pm 0.00$ |
| GOGGLE | $\mathbf{0.62 \pm 0.02}$ | $\mathbf{0.69 \pm 0.03}$ | $\mathbf{-0.01 \pm 0.00}$ | $\mathbf{0.60 \pm 0.06}$ | $\mathbf{0.70 \pm 0.02}$ | $\mathbf{-0.04 \pm 0.00}$ |

Figure 3: **Prior knowledge and generation.** ECOLI **(top row)** and MAGIC **(bottom row)**.

model—to the best of our knowledge, it is the first generative model that can leverage prior knowledge to generate more realistic synthetic samples.

## 5.3 ABLATION STUDY

GOGGLE is designed with the joint learning of both graph and functional relationships model. Having empirically demonstrated strong overall results, an immediate question is how important the dynamics of the two parts are for performance. Specifically, we consider the performance gain due to joint learning over learning the two parts separately. This includes the case where a graph is first learned separately: (1) sparse graph initialized using a sparse Erdos-Renyi random graph, with $10\%$ connectivity (**GOGGLE-ER**) [18], (2) initialized using cross-correlation matrix between variables (**GOGGLE-COV**), (3) learned using the PC algorithm (**GOGGLE-BN**) [64]. We also consider the case (4) when a graph is not learned but initialized as a fully connected dense graph (**GOGGLE-DENSE**); and (5) when the functional relationships model is replaced by a predictor that generates each variable using its neighborhood (**GOGGLE-NMP**). We perform an ablation study to report the performance achieved by each of these settings in Table 4.

We empirically observe that, the joint learning of both components is crucial to achieve consistently good performance. More specifically, we note that GOGGLE-ER achieves the worst performance across all settings. This is expected as the graph is randomly initialized. GOGGLE-COV learns an undirected graph, and only considers first order dependencies, leading to worse performance, while GOGGLE-BN learns a directed graph, which is too restrictive an inductive bias if incorrect. GOGGLE-DENSE employs a fully connected graph, and is equivalent to MLP-based generative models. While it achieved lower detection on Credit, it is prone to overfit, resulting in lower quality and utility scores. Lastly, GOGGLE-NMP generates each variable using only its immediate neighbors and ignores information from variables more than *1*-hop away, leading to worse performance.

## 6 DISCUSSION

In summary, we proposed GOGGLE, a novel tabular data generative model that jointly learns the relational structure and functional relationships through a message passing scheme to better capture the sparsity and heterogeneous relationships in tabular data. The explicit use of relational structure to guide generation allows prior knowledge and regularization on variable dependencies to be directly modelled. **Limitations and future work.** In this work, we focus on tabular data regime, where $d < 100$, future works can focus on higher-dimensional tabular data (e.g. genomics). To scale to higher-dimensions, we suggest extending the relational structure learning to the representation space for better generative models [43].

## ACKNOWLEDGMENTS

We thank the anonymous ICLR reviewers as well as members of the van der Schaar lab for many insightful comments and suggestions. Tennison Liu would like to thank AstraZeneca for their sponsorship and support. Jeroen Berrevoets thanks W.D. Armstrong Trust for their support. This work is also supported by the National Science Foundation (NSF, grant number 1722516) and the Office of Naval Research (ONR).

## ETHICS AND REPRODUCIBILITY STATEMENT

**Ethics.** This paper addresses a fundamental and well-established problem in ML. Synthetic data has the potential for social good, by allowing unfettered access to otherwise private data, an aspect which should be explored in future works. However, we provide a word of caution as synthetic data will inherit potential bias (if any) in the original dataset and inclusion of incorrect prior knowledge can lead to unintended downstream effects.

**Reproducibility.** We detailed exact implementation details, including dataset preprocessing, implementation of benchmark methods, architecture design, hyperparameter tuning, and evaluation methods in Section 4, Section 5, and Appendix B. All datasets used in this work can be downloaded from the UCI repository [16] and we provide the code to produce our results at https://github.com/tennisonliu/GOGGLE and the wider lab repository https://github.com/vanderschaarlab/GOGGLE.

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

## A    DATA AUGMENTATION MODEL

The data augmentation model conditionally generates augmented data, i.e. $\hat{x} = F_\theta(z, X_i = x_i; G_\phi)$. To allow conditioning on any variable during the generation stage, we assume a directed acyclic graph (DAG). This means that variables are generated sequentially, following topological order $\pi_{G_\phi}$, which can be described through a SEM [50]:

$$\hat{x}_i = f_i(\hat{\mathcal{N}}(i), z_i)$$

In contrast to the synthetic data model, the generation of each variable is conditioned on the predicted value of its parents. [5] Each variable is transformed through a specific embedding function $h_i^{(0)} = f_{\theta_i}(\hat{x}_i) \,\forall\, i \in [d]$, where $h_i^{(0)} \in \mathbb{R}^{d_0}$ and we implement $f_{\theta_i}$ using a one layer MLP. This is followed by one round of message passing, where the embedding update function is implemented using a single-layer MLP. We aggregate parent embeddings using the mean, equivalent to taking a weighted average:

$$m_j^{(1)} = \sigma_\theta^{(1)}(\cdot) = h_j^{(0)}$$
$$h_{\mathcal{N}(i)}^{(1)} = \oplus^{(1)}(\cdot) = \text{mean}\left(\{G_{j,i} m_j^{(1)}, \,\forall\, j \in \mathcal{N}(i)\}\right)$$
$$\hat{x}_i = \gamma_\theta^{(1)}(\cdot) = \tanh\left(W_u^{(1)} \times h_{\mathcal{N}(i)}^{(1)}\right)$$

The output after message passing is taken to be the prediction of the current variable $\hat{x}_i$. This model is trained using the GAN adversarial loss (eq. (5)). Additionally, we use the continuous DAG penalty introduced in [84] to force a DAG to be recovered:

$$\max_{F_\theta} \min_D \underbrace{\mathbb{E}[\log D(F_\theta(Z; G_\phi)) + \log(1 - D(X))]}_{GAN Loss} + \underbrace{\lambda(\text{tr}(\exp(G_\phi \circ G_\phi)) - d)}_{Regularisation} \qquad (5)$$

To generate augmented data, we randomly sample a subset of the variables, and for each of which, we sample a conditioning value *uniformly* from the support of that variable, $a \sim \mathcal{U}(\min(X_i), \max(X_i))$. We then generate augmented data by conditioning chosen variables on sampled values, $\hat{x} \sim p(X|X_i = a) = F_\theta(z, X_i = a; G_\phi)$ [50]. We note that this conditional generation corresponds to sampling *out-of-distribution*, as we are sampling from a different distribution that is defined on the same support as the training.

## B    IMPLEMENTATION DETAILS

### B.1    MODELS AND EVALUATION

**Models.** All models are implemented in PyTorch [48]. The data is split 60-20-20 into train, validation and test sets and reported results are averaged over 10 runs. Training is performed using the Adam [34]. All experiments are run on an NVIDIA Tesla K40C GPU, taking less than an hour to complete.

**Hyperparameters.** For all methods compared, we consider hyperparameters include batch size $\in \{64, 128\}$, learning rate $\in \{1e{-}3, 5e{-}3, 1e{-}2\}$. We include a weight decay of $1e{-}3$ [58]. Hyperparameters are searched using Bayesian Optimization [63]. Each model is allowed a computation budget of 10 sweeps in Bayesian Optimization, where the search objective is the reconstruction loss on the validation set. We describe model-specific hyperparameters in the exact implementation details below. For the graph sparsity term, we consider regularization penalty $\lambda \in \{1e{-}3, 1e{-}2, 1e{-}1\}$. For the KL divergence penalty, we consider $\alpha \in \{0.1, 0.5, 1.0\}$. All models are trained for a maximum of 1000 epochs, with early stopping if no improvements on the validation set for 50 epochs.

**Evaluation.** We partition the observed dataset $\mathcal{D}$ into a training set $\mathcal{D}_{train}$ and a test set $\mathcal{D}_{test}$, and train generative models on $\mathcal{D}_{train}$. Using the trained models, we generate a synthetic data set $\mathcal{D}_{syn}$, which has the same number of samples as $\mathcal{D}_{test}$. We then evaluate the aforementioned desiderata on $(\mathcal{D}_{test}, \mathcal{D}_{syn})$. To evaluate **quality**, we employ the three-dimensional metric, $\alpha$-precision, $\beta$-recall, and authenticity, which assesses whether the samples are realistic, diverse enough to cover the

---

[5]$\mathcal{N}(i) = Pa(i)$, the neighborhood is strictly the set of parent variables due to the DAG.

variability in real data, and generalization performance respectively [1]. Each metric evaluates a different aspect of synthetic data quality, and we average the three metrics to obtain a holistic score. **Detection** is evaluated by training post-hoc classifiers to distinguish samples from the original and generated datasets. Specifically, we train a two-layer MLP, XGB classifier and GMM classifier and average the classification AUROC. To evaluate **utility**, we report average performance achieved by three downstream prediction models (linear model, two-layer MLP, and XGB model) trained on $\mathcal{D}_{syn}$ and evaluated on $\mathcal{D}_{test}$. We report the change in AUROC of models trained on $\mathcal{D}_{syn}$ and those trained on $\mathcal{D}_{train}$.

## B.2 SYNTHETIC DATA BENCHMARKS

In this subsection, we provide further details on the benchmarks we compare against: including Bayesian networks (BN) [51], GAN-based [76], TableGAN [47] and VAE-based [76, 49] and a normalizing flow NFLOW [55]. Additionally, we also consider data augmentation methods: Gaussian noise InNoise [40]; MixUp [83] and SwapNoise [30]. The baselines are implemented using the open source package synthcity [54].[6]

**MM [53].** We train a Gaussian mixture model, where the number of components $\in \{5, 10, 15\}$, and each component has its own general covariance matrix. We use Expectation-Maximization to fit the parameters of the model, with a stopping condition at $1e$-3 or max iteration of 200 iterations. Lastly, we initialize the model using $k$-means clustering.

**BN [51].** We train BN in two stages, where the first stage learns the network structure and the second stage performs learning based on the returned DAG. We use the PC algorithm to learn the DAG [64]. Once a DAG is returned, the conditional probabilities are learned through maximum likelihood estimation, where continuous variables are assumed to come from a linear Gaussian conditional probability distribution (CPD) and discrete variables from a discrete CPD.

**CTGAN [76], TableGAN [47].** For CTGAN, we use an MLP with two ReLU-activated hidden layers to implement the generator. Similarly, we employ an MLP with two ReLU-activated hidden layers to implement the discriminator. The hyperparameters are tuned according to the recommended settings in [49]. TableGAN is implemented using a Deep Convolution GAN with recommended settings in [47], where the generator has three deconvolutional layers, and the discriminator has three convolutional layers.

**TVAE [49].** The VAE-based model is implemented with an encoder with two ReLU-activated layers. The decoder similarly has two hidden layers. We use a 32 dimensional latent space that is normally distributed and a standard normal prior.

**NFLOW [55].** We implement the normalizing flows using the rational-quadratic transform introduced in [17]. Specifically, it is implemented using an MLP with 2, 128-dimensional hidden layers and permutation operations. A standard normal base distribution is employed, and the flow is run with $500$ steps.

## B.3 DATA AUGMENTATION BENCHMARKS

**InNoise [40], MixUp [83], SwapNoise [30].** For InNoise, we add zero-centered Gaussian noise $\varepsilon \sim N(0, \sigma^2)$ to the inputs, where we consider $\sigma \in \{0.01, 1\}$. For SwapNoise, we randomly swap $10\%$ of elements between two inputs. MixUp is implemented by randomly combining two samples $\hat{x} = \lambda x' + (1 - \lambda)x''$, where $x', x'' \sim \mathcal{D}_{train}$ and $\lambda \sim \text{Beta}(0.2, 0.2)$.

**DAFS [14].** We train an autoencoder, where the encoder and decoder are both implemented as MLPs with two ReLU-activated hidden layers. We take the feature vector at the output of the encoder $c_i$ and randomly apply one of three possible operations (1) add Gaussian noise $\varepsilon \sim (0, \sigma^2)$, (2) interpolation $\lambda(c_j - c_i) + c_i$, or (3) extrapolation $\lambda(c_i - c_j) + c_i$ where $\lambda = 0.5$ as suggested by the authors. Augmented samples $x'$ are then obtained by passing the altered feature vector through the decoder.

---

[6]https://synthcity.readthedocs.io/en/latest/

Table 5: **Experimental datasets.** Description of experimental datasets.

| Dataset | Description | Number of instances | Number of features |
|---|---|---|---|
| Adult [38] | Census data | 48842 | 15 |
| Breast [67] | Breast cancer | 569 | 32 |
| Covertype [5] | Forest cover | 581012 | 54 |
| Credit [28] | Credit risk | 1000 | 20 |
| ECOLI [57] | Functional genomics | 2000 | 46 |
| MAGIC-IRRI [59] | Plant genetics | 2000 | 64 |
| Red [12] | Wine quality | 1599 | 12 |
| White [12] | Wine quality | 4898 | 12 |
| Mice [26] | Protein expression | 1080 | 82 |
| Musk [16] | Musk molecules | 6598 | 168 |

## B.4 DATASETS

We use 10 datasets in total, including 8 UCI datasets [16],[7] specifically `Adult`, `Breast`, `Covertype`, `Credit`, `White`, `Red`, `Mice`, `Musk` and 2 Bayesian Network repository datasets [39],[8] specifically `ECOLI` and `MAGIC-IRRI`. A summary of the datasets, including dataset description, the dimensionality, and number of samples, is presented in Table 5.

## B.5 PRIOR KNOWLEDGE

The use of an explicit graph to guide generation allows for a variety of prior knowledge to be incorporated through the adjacency matrix. Here, we describe a few options of incorporating domain expertise:

- **Sparsity.** The learned graph can be sparse such that variables only depend on a small subset of other variables. Mathematically, $\mathcal{R}(G_\phi) = ||G_\phi||_p$, where $|| \cdot ||_p$ denotes the $L_p$ matrix norm.
- **Dependence.** Partial knowledge about the dependencies between features can be encoded through a graph prior $G_0$, i.e., $\mathcal{R}(G_\phi) = ||G_\phi - G_0||_p$.
- **Graph types.** Graphs of specific types can be learned. For example, if an undirected graph is assumed, we can employ a symmetric prior $||G_\phi - G_\phi^T||_p$. Alternatively, we can use the DAG penalty [84] to encourage learning a directed, acyclic graph (DAG), $\mathcal{R}(G_\phi) = \text{tr}(\exp(G_\phi \circ G_\phi) - D)$, where $tr(\cdot)$ is the matrix trace and $D$ is the number of variables.
- **Connectivity.** Encourage different patterns of connectivity through penalty on degree of each variable $||D_\phi - D_0||_p$, where $D \in \mathbb{R}^d$ is the degree of each variable.

## C ADDITIONAL EXPERIMENTS

In this section, we provide additional results to comprehensively evaluate our proposed methods, specifically:

1. **Additional datasets**: §C.1 evaluates synthetic data performance on 4 additional datasets.
2. **Visualizations**:§C.2 visualizes t-SNE projections on original and synthetic datasets to qualitatively investigate quality and examines learned adjacency matrices.
3. **Sensitivity**: §C.4 investigates performance sensitivities according to data size and feature counts.
4. **Data augmentation**: §C.5 describes the best model performance after data augmentation.

## C.1 ADDITIONAL RESULTS

**Additional datasets.** We assess the quality of synthetic dataset using the same desiderata introduced in §5.1, namely *quality*, *detection*, and *utility*. We use six additional datasets, `ECOLI`, `MAGIC-IRRI`, `Red`, `White`, `Mice`, and `Musk`. The results are reported in Table 6.

---

[7] https://archive.ics.uci.edu/ml/datasets.php

[8] https://www.bnlearn.com/bnrepository/

Table 6: **Quality, detection, and utility of synthetic data.** Bold indicates the best performance.

| | Dataset | ECOLI | MAGIC-IRRI | Red | White | Mice | Musk |
|---|---|---|---|---|---|---|---|
| **Quality (↑)** (higher is better) | BN | **0.57 ± 0.06** | **0.67 ± 0.04** | 0.63 ± 0.06 | 0.60 ± 0.04 | **0.63 ± 0.02** | 0.58 ± 0.05 |
| | CTGAN | 0.38 ± 0.06 | 0.33 ± 0.10 | 0.50 ± 0.07 | 0.53 ± 0.04 | 0.46 ± 0.05 | 0.48 ± 0.04 |
| | TableGAN | 0.38 ± 0.08 | 0.33 ± 0.07 | 0.46 ± 0.08 | 0.53 ± 0.02 | 0.41 ± 0.04 | 0.56 ± 0.03 |
| | TVAE | 0.45 ± 0.09 | 0.61 ± 0.02 | 0.57 ± 0.04 | 0.58 ± 0.05 | 0.57 ± 0.05 | 0.57 ± 0.04 |
| | NFLOW | 0.56 ± 0.08 | 0.62 ± 0.07 | 0.56 ± 0.03 | 0.55 ± 0.04 | 0.52 ± 0.07 | 0.57 ± 0.03 |
| | GOGGLE | **0.57 ± 0.05** | 0.63 ± 0.09 | **0.63 ± 0.07** | **0.62 ± 0.03** | 0.59 ± 0.05 | **0.61 ± 0.02** |
| **Detection (↓)** (lower is better) | BN | **0.39 ± 0.07** | **0.40 ± 0.03** | 0.71 ± 0.06 | **0.60 ± 0.03** | 0.73 ± 0.05 | 0.80 ± 0.05 |
| | CTGAN | 0.74 ± 0.09 | 0.73 ± 0.07 | 0.77 ± 0.10 | 0.81 ± 0.06 | 0.75 ± 0.03 | 0.75 ± 0.03 |
| | TableGAN | 0.74 ± 0.02 | 0.73 ± 0.06 | 0.74 ± 0.04 | 0.75 ± 0.09 | 0.80 ± 0.06 | 0.78 ± 0.05 |
| | TVAE | 0.74 ± 0.02 | 0.69 ± 0.05 | 0.72 ± 0.07 | 0.74 ± 0.04 | **0.71 ± 0.06** | 0.77 ± 0.06 |
| | NFLOW | 0.70 ± 0.03 | 0.70 ± 0.08 | 0.74 ± 0.05 | 0.73 ± 0.03 | **0.71 ± 0.03** | 0.73 ± 0.05 |
| | GOGGLE | 0.60 ± 0.03 | 0.69 ± 0.09 | 0.71 ± 0.04 | 0.70 ± 0.05 | 0.72 ± 0.04 | **0.69 ± 0.08** |
| **Utility (↑)** ( higher is better) | BN | **0.01 ± 0.00** | **0.05 ± 0.00** | −0.06±0.01 | −0.11±0.04 | **−0.02 ± 0.00** | −0.19±0.06 |
| | CTGAN | −0.20±0.03 | −0.13±0.01 | **0.01 ± 0.00** | −0.11±0.02 | −0.08±0.03 | −0.13±0.04 |
| | TableGAN | −0.18±0.06 | −0.10±0.05 | **0.01 ± 0.00** | −0.17±0.01 | −0.15±0.04 | −0.10±0.03 |
| | TVAE | −0.06±0.01 | 0.00 ± 0.00 | −0.05±0.01 | −0.11±0.02 | −0.09±0.02 | **−0.08 ± 0.01** |
| | NFLOW | −0.05±0.01 | −0.02±0.00 | −0.05±0.01 | −0.14±0.05 | −0.08±0.02 | −0.14±0.05 |
| | GOGGLE | −0.02±0.00 | 0.01 ± 0.00 | **0.01 ± 0.00** | **−0.08 ± 0.01** | −0.10±0.02 | −0.11±0.03 |

We note that BN achieves the best performance on ECOLI and MAGIC-IRRI, which is reasonable as those datasets are generated according to a known Bayesian network, and BN models have a natural advantage. On those two datasets, GOGGLE is able to consistently outperform other deep generative models. On Red and White, GOGGLE achieves superior performance against other benchmarks. On the contrary, BN, our closest competitor, achieve worse performance as the underlying DAG assumptions become too restrictive. Additionally, we highlight that models trained on synthetic data generated by GOGGLE consistently achieves similar performance to those trained on real datasets, indicating strong data utility.

## C.2 VISUALIZATION OF SYNTHETIC DATA RESULTS

In Figures 4 to 7, we observe that synthetic data generated by GOGGLE exhibit markedly better overlap with the original dataset than other benchmarks using t-SNE for visualization. We note that the GAN-based models, specifically CTGAN and TableGAN exhibit mode collapse behaviour and the TVAE and NFLOW can fail to match the underlying distribution (on ECOLI and Breast, respectively).

## C.3 ANALYSIS OF RELATIONAL STRUCTURES

**Qualitative analysis.** We visualize the learned graphs on Credit and Breast in Figure 9. For the purposes of our qualitative analysis, we deliberately increase the weighting of the graph sparsity regularization $\lambda$. The Breast dataset [16] contains numeric features extracted from images of a breast mass. We note that the *target* variable (diagnosis of tumor) has a high degree of connectivity, and dependent on various physical properties of the tumor, including *mean perimeter* and *mean compactness*. We similarly observe informative variables identified in the Credit dataset [16], where the *account balance* depends on *occupation*, *credit amount*, and *length of current employment*. Additionally, we plot the adjacency matrix of trained models in Figure 8, where a sparsity regularization term was applied to all models to encourage sparsely connected graphs.

**Quantitative analysis.** We previously claimed that we are only interested in an *approximately correct* relational structure, which our evaluations in Table 4 and Figure 3 found is sufficient to improve synthetic data quality. However, are these structures indeed approximately correct? That is the question we aim to address here. We compute the structural hamming distance (SHD) [13], which computes graph distances between predicted and ground truth graphs by the number of insertions, deletions or flips to transform one graph to another. We compare the relational structure learned by GOGGLE against 1) the graph learned by a Bayesian Network (BN), 2) Erdos-Renyi random graph generated with edge probability matching empirical edge probability $p$ in the ground-truth graph (ER(p)), 3) thresholded correlation graph (CORR).

We describe results in Table 7. We note that BN model is the *pseudo*-oracle as the true DGP for ECOLI and MAGIC are indeed Bayesian Networks with linear Gaussian functional relationships (matching the model specifications of BN). ER(p) serves as a dummy baseline as the predicted graph is randomly guessed. We note that the correlation graphs (CORR, with threshold at 0.5) learn many

Table 7: **Structural hamming distance.** Distance from learned graph to ground truth graph.

| Datasets | BN | ER(p) | CORR | GOGGLE |
|---|---|---|---|---|
| **ECOLI** | $61 \pm 2$ | $131 \pm 6$ | 204 | $110 \pm 5$ |
| **MAGIC** | $82 \pm 3$ | $192 \pm 10$ | 103 | $95 \pm 2$ |

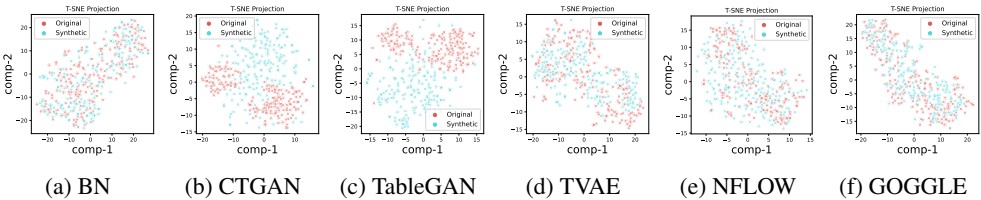

(a) BN    (b) CTGAN    (c) TableGAN    (d) TVAE    (e) NFLOW    (f) GOGGLE

Figure 4: **t-SNE projection on Breast dataset.**

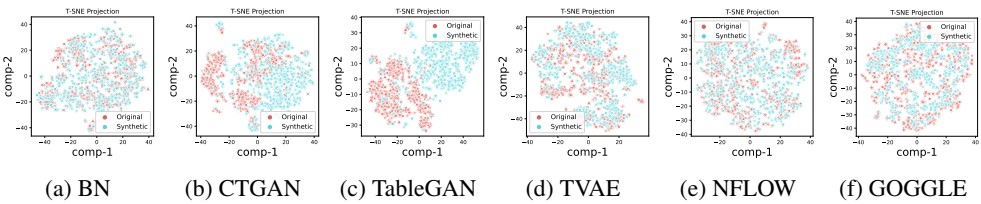

(a) BN    (b) CTGAN    (c) TableGAN    (d) TVAE    (e) NFLOW    (f) GOGGLE

Figure 5: **t-SNE projection on Red dataset.**

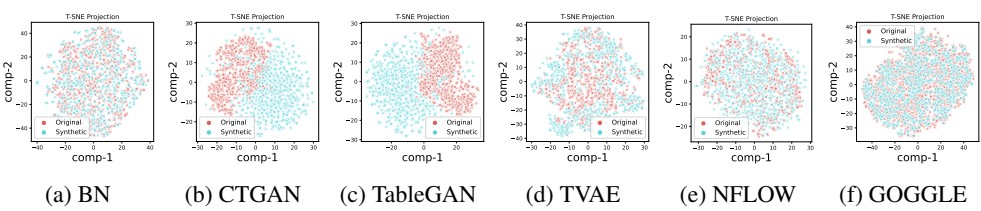

(a) BN    (b) CTGAN    (c) TableGAN    (d) TVAE    (e) NFLOW    (f) GOGGLE

Figure 6: **t-SNE projection on ECOLI dataset.**

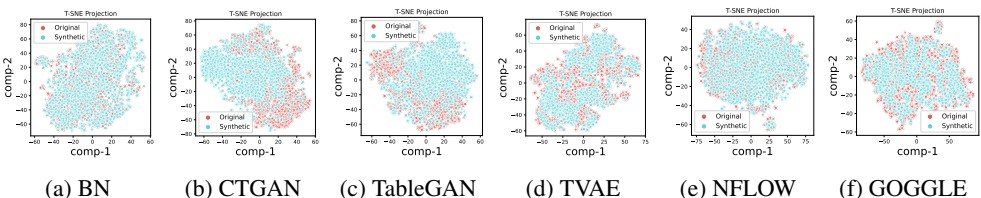

(a) BN    (b) CTGAN    (c) TableGAN    (d) TVAE    (e) NFLOW    (f) GOGGLE

Figure 7: **t-SNE projection on White dataset.**

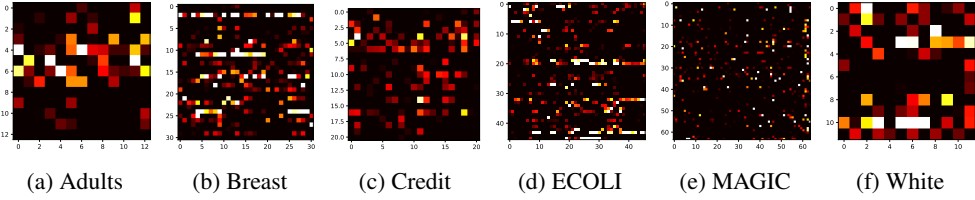

(a) Adults    (b) Breast    (c) Credit    (d) ECOLI    (e) MAGIC    (f) White

Figure 8: **Learned adjacency matrices.**

spurious, yet uninformative edges as it is solely driven by association relationships. We note that the graphs learned by GOGGLE are more sparse, and informative.

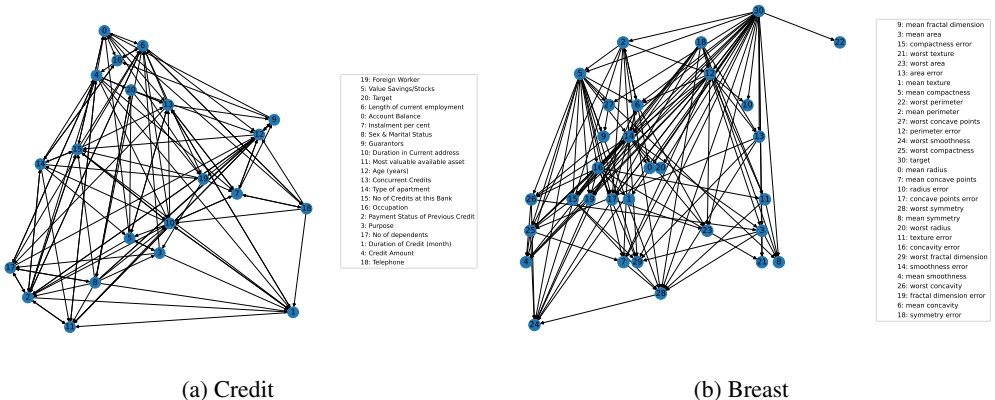

(a) Credit          (b) Breast

Figure 9: **Learned graphs.**

Table 8: **Data augmentation**. AUROC on $\mathcal{D}_{test}$ of models trained on augmented data. Bold indicates the best performance.

| Dataset | Adult | Breast | Covertype | Credit |
|---|---|---|---|---|
| Baseline | $0.70 \pm 0.09$ | $0.96 \pm 0.01$ | $0.59 \pm 0.13$ | $0.65 \pm 0.02$ |
| InNoise | $0.68 \pm 0.05$ | $0.97 \pm 0.01$ | $0.61 \pm 0.10$ | $0.64 \pm 0.03$ |
| MixUp | $0.66 \pm 0.08$ | $0.90 \pm 0.01$ | $0.56 \pm 0.09$ | $0.65 \pm 0.02$ |
| SwapNoise | $0.65 \pm 0.05$ | $0.91 \pm 0.01$ | $0.58 \pm 0.08$ | $0.63 \pm 0.01$ |
| FSAug | $0.71 \pm 0.06$ | $0.96 \pm 0.01$ | $0.60 \pm 0.13$ | $\mathbf{0.67 \pm 0.02}$ |
| GOGGLE-SD | $0.70 \pm 0.05$ | $0.97 \pm 0.01$ | $0.60 \pm 0.10$ | $0.66 \pm 0.01$ |
| GOGGLE | $\mathbf{0.72 \pm 0.03}$ | $\mathbf{0.98 \pm 0.00}$ | $\mathbf{0.62 \pm 0.10}$ | $\mathbf{0.67 \pm 0.02}$ |

## C.4 SENSITIVITY ANALYSIS

Lastly, we are interested in identifying settings where GOGGLE excel as a generative model. Specifically, we are interested in understanding sensitivities of model performance with respect to the effects of feature counts and number of samples in the dataset. We compare our model against the benchmarks in Figure 10. Here, the datasets are shown on the x-axis and are sorted in order of increasing feature count, and increasing number of samples (see Table 5 for more on datasets). We note that the advantage of GOGGLE is more noticeable when there are less number of samples (i.e., on Breast, Credit and Red). In the regime with larger number of samples, all models exhibit similar performance, although GOGGLE still achieves performance improvements. Furthermore, models achieve similar performance when the number of features is low. However, when the number of features increases, the performance of GAN-based models deteriorate. This is interesting, and a potential logical explanation is that they are overfitting to the training data.

## C.5 EVALUATION OF AUGMENTED DATA

We compare popular tabular data augmentation methods by inspecting downstream model performance. Specifically, we generate augmented data $\mathcal{D}_{aug}$ from $\mathcal{D}_{train}$ (and has the same number of samples), train predictive models on the combined $\mathcal{D}_{comb} = \{\mathcal{D}_{train}, \mathcal{D}_{aug}\}$, and evaluate performance on $\mathcal{D}_{test}$. We train four downstream prediction models, including linear model, two-layer MLP, RF classifier, and XGB model and report the averaged performance achieved by the four models. We perform data augmentation on GOGGLE by randomly selecting variables to condition on and sampling uniformly from the marginal support of the variable. In Table 8, we observe that augmented data generated by GOGGLE leads to improved generalization performance across all datasets.

## D CONNECTION TO RELATED WORKS

There are several parallels between our works and several related research fields, namely *probabilistic graph discovery*, *relational learning*, and *self-supervised learning* (SSL). These fields are reflected by the common approach in exploiting relational structure underlying tabular data. Here, we discuss commonalities and differences in depth.

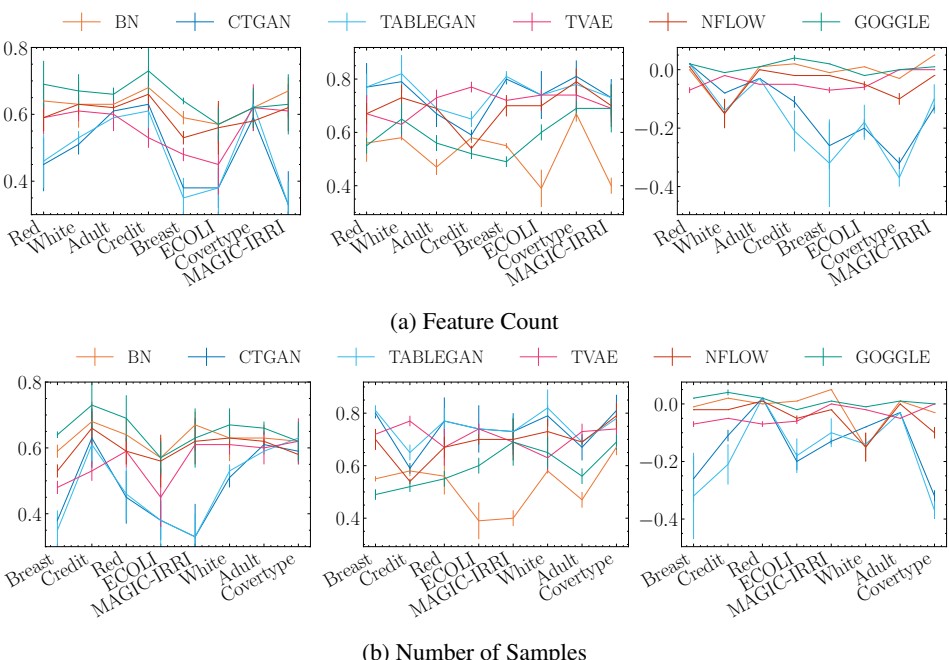

(a) Feature Count

(b) Number of Samples

Figure 10: **Sensitivity analysis.** Evaluating synthetic data based on **(left) quality** ($\uparrow$), **(middle) detection** ($\downarrow$), and **(right) utility** ($\uparrow$). Datasets are sorted according to **(a)** increasing feature counts, and **(b)** increasing number of samples.

**Probabilistic graph discovery.** Probabilistic graph discovery aims to recover the true probabilistic graphical model (PGM) underlying observed data [85, 15]. We propose a generative model for tabular data that learns and leverages an underlying graph to improve the performance of data synthesis. Importantly, our model does not recover the true PGM from data if the data is indeed generated by a PGM (i.e. it does not perform PGM structural discovery). Additionally, the message passing computation is not an instance of or an approximation to a probabilistic inference routine. Specifically, we take advantage of the sparse and compact representations of graphical models to learn better generative models, incorporate prior knowledge, and perform conditional generation. We do so by incorporating a graph as explicit structure into the generative process. We summarize the key distinctions between our work and the probabilistic structure learning literature.

The graph learned in GOGGLE encodes conditional dependence structure between variables (global Markov property), in the same sense that PGMs reflect allowed conditional dependencies. The key distinction between our approach and probabilistic structural learners is that we only require an approximate structure. In contrast, structural learners aim to recover unique graphs that are close to the true DGP. This learned graph is used to answer probabilistic inference queries, e.g. $P(Y|X = x)$, which requires the graph to be correct. In order to recover a unique graph, structural learns generally assume certain graph, or distributions, or relationships between variables.

Our objective is to learn an approximate graph that models associational dependence and can guide generation. Therefore, we do not need to make similar assumptions that unnecessarily restrict the class of learnable distributions and can lead to a miss-specified model. Additionally, we emphasize that our proposed method is not designed to perform sampling-based probabilistic inference. Due to the different objectives, we make minimal assumptions on the *graph type*, *variable distribution*, and *functional relations*.

**Relational learning.** The goal of relational inference is to infer relationships between objects from observational data alone (interacting objects in NRI [36], and molecular interactions in multi-omics integration in BayRel [24] and MoReL [25]). The output of the learning method is a probabilistic relational graph, which is evaluated based on correctness of the recovered graph. The biggest difference of our work is that GOGGLE is fine if the structure is only *partially correct*, or wrong, which is in stark contrast to any literature on structure discovery. The relational structure in our case

Table 9: **Comparison to related works.** Commonalities and differences between our work, relational inference, and probabilistic structural discovery.

| | Probabilistic Graph Discovery | GOGGLE |
|---|---|---|
| **Commonality** | Edges reflect allowed conditional dependencies between variables | |
| **Objective** | Recover unique probabilistic graph underlying observed data | Learn approximate graph describing dependencies between variables |
| **Evaluation Metric** | Quality of discovered graph: graph distance measure [52, 61], edge classification metric | Quality of synthetic data: quality, detection, and utility [1] |
| **Application** | Probabilistic inference $P(Y\|X = x)$ | Generate conditional synthetic data $x \sim P_\theta(X)$ |
| **Assumptions** | Specific graph types (i.e. directed or undirected) | Arbitrary graph types (i.e. mixed, directed, or undirected) |
| | Distributional assumptions on variables (e.g. Gaussian) | No assumptions on variable distribution |
| | Assumptions on functional relationships between variables (e.g. Linear with Gaussian additive noise) | No assumptions on functional relationships model |
| **Representative Works** | **UGM**: Chow-Liu algorithm [11], graphical LASSO [2], neighborhood selection [44]. **DGM**: score-based [9], constraint-based [65], hybrid [70] | **Deep generative**: GAN-based [76], VAE-based [77]. **Non-neural**: BN, mixture models, copula |

acts as an inductive bias encouraging a sparse set of informative neighbors to be found, thus better learning the distribution of tabular data, and plays a regularization effect on spurious relationships. As we show in Table 4 and Figure 3, these partially correct structures that we learn can consistently improve generative synthetic data performance.

**SSL.** SSL methods for tabular data are similarly driven by uncovering relational information between features to learn good representations. However, this is often done implicitly, as in VIME [81], which applies a masking operator to encourage the representation learning module to learn inter-feature relationships. Similarly, SubTab [71] employs a contrastive loss by generating views on subsets of features, implicitly encouraging learned representations to extract mutual information. Our work is different in explicitly incorporating a relational mechanism into the generative process, explicitly encouraging informative relations to be found, in addition to providing a flexible mechanism for prior knowledge and regularization. We discuss specific differences in objectives, methods and evaluation criteria below:

Table 10: **Comparison to related works.** Commonalities and differences between our work, relational inference, and self-supervised learning (SSL).

| | Relational Inference | SSL | GOGGLE |
|---|---|---|---|
| **Commonality** | Exploit relational structure between variables | | |
| **Learning output (goal)** | The relational graph $G$ | Representation vector $h$ | Data distribution $p(X)$ |
| **Learning method** | Probabilistic inference over relational graph $p(G\|X)$ | Learning encoder function $h = f(x)$ through contrastive loss or pretext generation | Joint learning of the relation graph $G$ and the distribution $p(X)$ that is compatible with $G$ |
| **Evaluation** | Edge prediction metric; graph distance metric | Quality of representations, e.g. performance in downstream tasks | Quality of synthetic data: quality, detection, utility |
| **Representative works** | NRI [36], BayReL [24], MoReL [25] | VIME [81], SubTab [71] | CT-GAN [77], TVAE [76] |

