# OpenReview forum: "GOGGLE: Generative Modelling for Tabular Data by Learning Relational Structure"
_ICLR.cc/2023/Conference — ICLR 2023 poster_

### Official Review · Reviewer_Hvkk · 2022-10-21

**Confidence:** 3
**Correctness:** 3
**Technical Novelty And Significance:** 4
**Empirical Novelty And Significance:** 3
**Recommendation:** 8

**Clarity, Quality, Novelty And Reproducibility:**

This is a very clear paper, the contribution seems novel and important.

Extensive description of the model and experimental setting is given in the paper and appendix. One notable exception is the value used for the $L$ hyperparameter in the various experiments that I could not find. The author plan to release the full source code if the paper is accepted.

**Strength And Weaknesses:**

**Strengths:**
- The proposed architecture is well justified and well detailed. The goals and non-goals of the model are clearly stated (notably appendix D is a very welcome comparison).
- Experiments are detailed and extensive, and the ablation study clearly illustrates the impact of the core contribution (the GraphNN structure of the decoder).
- The introduced model seems to have a good potential for meaningful impact on the problem of synthetic tabular data, as expressed by the performance impact, in particular in terms of *utility* (using the generated synthetic data for downstream tasks).

**Weaknesses:**
- The probabilistic part of the decoder is hardly discussed, while potentially having a big impact on the performance of the model. Appendix explains that a MSE loss is used for continuous variables, which corresponds to an isotropic Gaussian noise with fixed variance. See [*Simple and Effective VAE Training with Calibrated Decoders*, Rybkin et al, 2021](https://proceedings.mlr.press/v139/rybkin21a.html) for a discussion of why this is an important factor over the model performance. That paper focuses on image data, but the core argument translates to tabular data. I believe learning a per-feature noise scale for the Gaussian observation model may further improve the performance of the proposed model.
- The is no discussion of the impact of $L$, the number of GraphNN layers stacked in the decoder. I suspect this parameter might depend a lot on the dataset at hand, and in particular of how sparse or strong the correlations between the variables are?
- The discussion suggests extending the approach to learning graphs in the latent representation rather than between the observed variable. There is actually already some work in this vein existing (see for example [*Learning Latent Superstructures in Variational Autoencoders for Deep Multidimensional Clustering*, Li et al, 2019](https://arxiv.org/abs/1803.05206)) which would be worth discussing in the related works section of the paper.

**Summary Of The Paper:**

The paper proposes a new generative model for tabular data, as a VAE whose decoder structure is built around GraphNN layers.

The rationale for this structure is given in terms of inductive biases: on structured data like images or text, the successful models are built around architectures that take advantage of good inductive biases derived from the structure of the data. Tabular data lacks this general structure, and instead generally has a problem-dependent structure. Models like MLP don't have any of those inductive biases, explaining why they don't perform so well on tabular data.

The paper thus introduces GOGGLE, a VAE-like generative model whose decoder half is built using GraphNN layers that act akin a message-passing scheme on top of a graph defined over the features of the data. Similar to a probabilistic graphical model (while *not* being one), this structure reflects the dependencies between the variables as learned by the model. The paper proposed several regularization schemes over the learned adjacency matrix of this graph to reflect different inductive biases that can be injected in the model such as encouraging sparsity or injecting a partially known dependency structure. While the constructed graph is *not* a proper PGM, it structures the way in which the generative model processes the data.

The paper  provides experimental validation on several datasets and against multiple other models from the literature tailored to tabular data. Experiments show that GOGGLE reaches better performance across the board and following multiple metrics, including the quality of generated samples to train a downstream classifier on synthetic data. GOGGLE is notably competitive with Bayesian Networks, which remained the state of the art on many tabular problems, suggesting that the injected graphical structure is indeed core to performance on these problems.

This is further confirmed by an ablation study, which shows that the quality of the graph underlying the GraphNN decoder has a significant effect on the performance of GOGGLE, and that learning it jointly with the rest of the model using the proposed regularization is notably better than trying to infer it beforehand using another method.

**Summary Of The Review:**

This is a good paper that provides a meaningful contribution to the question of generative modelling on tabular data.

---

> ### Author Response · Authors · 2022-11-11
> **Response to Reviewer Hvkk (2/2)**
>
> ## Q3. Graphs in the latent space
> *“learning graphs in the latent representation… related works”*
>
> **Response**
>
> [1] introduces graph learning on latent variables, where the graph is assumed to be a tree structure. This tree structure is employed to perform inference on top-level latent variables, where each variable represents a different facet of clustering, and each discrete state that a variable takes on defines a cluster. Similar to our method, [1] iteratively learns the latent structure and the generative model with the purpose of performing clustering. We strongly believe that future works should investigate combining variants of [1] with the relational structure learning proposed in this work to better model higher-dimensional tabular datasets. Thank you for this suggestion, and we have revised the manuscript to discuss this related work.
>
> **References**
>
> [1] Li, X., Chen, Z., Poon, L.K. and Zhang, N.L., 2018. Learning latent superstructures in variational autoencoders for deep multidimensional clustering. arXiv preprint arXiv:1803.05206.
>
> ---
> We thank the reviewer for your help in improving our work. Please let us know if our latest changes have addressed your concerns, and if there is anything else you would like to see.

---

> ### Author Response · Authors · 2022-11-11
> **Response to Reviewer Hvkk (1/2)**
>
> Thank you very much for your helpful feedback and comments. We aim to address all the individual points in your review here, but please also see the revised manuscript for changes (highlighted in blue).
>
> ---
> ## Q1. Probabilistic decoder
> *“probabilistic part of the decoder is hardly discussed, while potentially having a big impact on the performance of the model”*
>
> **Response**
>
> Thank you for pointing this out. As you stated, we employ an **uncalibrated** decoding distribution, specifically a Gaussian distribution with a fixed variance, an approach that is adopted by most existing works (see [1-3] for a few). The suggested improvement is agnostic to the exact generative model used. Indeed, our method can be easily adapted to incorporate either a learnable variance, or variance that is computed analytically from training data (i.e. the optimal $\sigma$-VAE in [4]). We incorporated this recommended reference into our revised manuscript.
>
> **References**
>
> [1] Lee, A.X., Nagabandi, A., Abbeel, P. and Levine, S., 2020. Stochastic latent actor-critic: Deep reinforcement learning with a latent variable model. Advances in Neural Information Processing Systems, 33, pp.741-752.
>
> [2] Castrejon, L., Ballas, N. and Courville, A., 2019. Improved conditional vrnns for video prediction. In Proceedings of the IEEE/CVF International Conference on Computer Vision (pp. 7608-7617).
>
> [3] Babaeizadeh, M., Finn, C., Erhan, D., Campbell, R.H. and Levine, S., 2017. Stochastic variational video prediction. arXiv preprint arXiv:1710.11252.
>
> [4] Rybkin, O., Daniilidis, K. and Levine, S., 2021, July. Simple and effective VAE training with calibrated decoders. In International Conference on Machine Learning (pp. 9179-9189). PMLR.
>
> ---
> ## Q2. Impact of L
> *“discussion of the impact of L, the number of layers stacked in the decoder”*
>
> **18/11 NOTE: see [updated results](https://openreview.net/forum?id=fPVRcJqspu&noteId=i6tG7SMXoIa)**
>
> **Response**
>
> This is a great point. In our work, we found that using $L=2$ was a suitable setting for all datasets. However, there is a fundamental tradeoff in choices of $L$ - in one round (or layer) of message passing, each variable receives messages from variables one-hop away. Accordingly, $L=1$ means each variable is generated using only its direct neighbors, which can be a strong assumption that increases the difficulty of generative learning. The shallow network also loses expressiveness in learning more complex distributions. On the other hand, as $L$ increases, each variable will receive information about all other variables in the graph, thus losing the sparse, heterogeneous inductive bias injected by the relational structure. As the network gets deeper, enabling larger receptive fields, performance of GNNs also deteriorates due to the over-smoothing issue [1-3].
>
> To further investigate the effect of $L$ on the performance of our model, we perform a sensitivity analysis based on simulations of tabular datasets with 1) different levels of correlation and 2) different levels of heterogeneity (or complexity of functional relationships). We are currently collecting results for this experiment, which will be submitted for your review as soon as they are ready.
>
> **Actions taken**
>
> **[pending]** Include additional sensitivity analysis on effect of $L$ on synthetic data performance using simulated datasets with different levels of correlation and heterogeneity.
>
> **References**
>
> [1] Liu, M., Gao, H. and Ji, S., 2020, August. Towards deeper graph neural networks. In Proceedings of the 26th ACM SIGKDD international conference on knowledge discovery & data mining (pp. 338-348).
>
> [2] Oono, K. and Suzuki, T., 2019, September. Graph Neural Networks Exponentially Lose Expressive Power for Node Classification. In International Conference on Learning Representations.
>
> [3] Zhao, L. and Akoglu, L., 2019, September. PairNorm: Tackling Oversmoothing in GNNs. In International Conference on Learning Representations.

---

> ### Author Response · Authors · 2022-11-19
> **Additional Results for Q2**
>
> Dear Reviewer Hvkk,
>
> Once again, **thank you** for your thoughtful review. We provide additional results for our previous comment titled *Q2. Impact of L*. Here, we added results for a sensitivity analysis of the impact of $L$ on synthetic data performance.
>
> ---
>
> To further understand the effect of $L$ on the performance, we perform a sensitivity analysis based on simulations of tabular datasets with 1) different levels of correlation and 2) different levels of heterogeneity (or complexity of functional relationships).
>
> **Correlation Simulation** We start by simulating a Erdos-Renyi random *undirected* graph $\mathbf{G}\sim ER(n, p)$ with $n$ nodes and $p$ probability for edge creation. Specifically, $\mathbf{G}$ is a $n\times n$ symmetric adjacency matrix where edges have $p$ probability of existing. We then sample a random correlation matrix $C\sim W(\mathbf{V}, n)$ from the Wishart distribution, where the $\mathbf{W}$ is the scale matrix that reflects the desired covariance structure. Specifically, $\mathbf{W} = \mathbf{I} + (\mathbf{1}-\mathbf{I})\cdot\rho$, where $\mathbf{I}$ is the $n \times n$ identity matrix and $\rho$ is the desired correlation. Lastly, we obtain a sparse covariance matrix $\mathbf{C}'$ by taking the Hadamard product between $\mathbf{C}' = \mathbf{C} \odot \mathbf{G}$. We then generate simulated data from the multivariate normal distribution with zero mean and the sparse covariance matrix $\mathcal{N}(\mathbf{0},\mathbf{C}')$.
>
> Specifically, we consider $n=20$ and $p=0.5$ and test sensitivities with respect to $\rho \in $ {0.1, 0.5, 0.9}. We train GOGGLE on simulated data and evaluate generated synthetic data in the results below.
>
> [[Additional results] Sensitivity of $L$ to increasing correlation levels.](https://i.imgur.com/yfJi1rc.png)
>
> We first note that, in line with what we observed in prior experiments, $L=2$ is a suitable setting for all correlation levels. At lower correlations, where very little information is shared between neighbours, shallow $L=1$ networks is sufficient to generate high quality synthetic data. However, as correlation increases, $L=2$ outperforms shallow networks as it better propagates shared information. $L=3$ suffers from lowest performance, likely due to label-smoothing, as the variables lose variable-specific information.
>
> **Complexity Simulation** We generate random *directed, acyclic graphs* (DAG) $\mathbf{G}\sim ER_{DAG}(n, p)$ with $n$ nodes and $p$ probability for edge creation. Note that we sample a DAG so we can control the complexity of the functional relationships. From the sampled graphs, we obtain a topological order $\pi_\mathbf{G}$, which is used to generate variables sequentially, i.e. $\hat{x}$\_i$ = f_i(\hat{\mathcal{N}}(i), z_i)$, where ${\mathcal{N}}(i)$ are the parent nodes of $x_i$ in $\mathbf{G}$. To simulate different complexities of $f_i$, we compute $\hat{x}$\_i = $\alpha \cdot f_{DL, i}(\hat{\mathcal{N}}(i))  + (1-\alpha)\cdot f_{linear, i}(\hat{\mathcal{N}}(i)) \;\forall\; i$, where $f_{linear, \cdot}$ and $f_{DL, \cdot}$ correspond to a randomly initialised linear function and non-linear, two-hidden layer MLP respectively. Varying $\alpha$ smoothly varies the complexity of the functional relations - at $\alpha=0$, the functional relations are simpler, linear functions, and at $\alpha=1$, the functional relations are complex non-linear functions.
>
> As before, we consider $n=20$ and $p=0.5$ and test sensitivities with respect to $\alpha \in $ {0, 0.25, 0.5, 0.75, 1}. We train GOGGLE on simulated data and evaluate generated synthetic data in the results below.
>
> [[Additional results] Sensitivity of $L$ to increasing complexity levels.](https://i.imgur.com/2k0mbpq.png)
>
> Similarly, at lower levels of relational complexity (i.e. $\alpha=0$), we see that $L=1$ is sufficient in learning the data distribution, but deeper, more expressive networks are better performing when complexity increases.
>
> We hope that the new set of results addresses your comment on effect of $L$, although please let us know if you still have concerns.

---

### Official Review · Reviewer_bxj6 · 2022-10-24

**Confidence:** 5
**Clarity, Quality, Novelty And Reproducibility:** Novelty is limited. And the reproduci…
**Correctness:** 3
**Technical Novelty And Significance:** 1
**Empirical Novelty And Significance:** 2
**Recommendation:** 3

**Strength And Weaknesses:**

**Pros:**

- Generative modeling for tabular data is a hard and interesting problem.

**Cons:**

- The novelty is limited.

- The authors missed some related papers, especially in the methodological parts, including NRI [ICML 2018], MoReL [ICLR 2022], BayReL [ NeurIPS 2020], and many other related relational inference papers in other domains.

- Some of the self-supervised learning models for tabular data can be used as generative models, including VIME [NeurIPS 2020], SubTab [NeurIPS 2021], and many other available methods which are based on self-supervised methods.

- The learned structure is not well explored in experiments. How it will look like and how is different from the ground truth?

- The datasets are not high dimensional.

- The authors did not report the performance of the raw data model without generated dataset.

- The authors missed some of the basic regression methods and is not clear how the reported ones are tuned.

- The authors reported an average of three different methods. I would like to see the performance of each separately, and also LR and simple methods should be included.

- The AUROC metrics is not sufficient. I would like to see the ROC plot as well as other metrics such as recall, FDR, F1, etc.

**Summary Of The Paper:**

The authors propose to use relational inference for the generative model in a tabular setting. The proposed method jointly learns the relation between features in tabular data and parameters of generative models.

The paper is fairly well structured, apart from missing some of the related works and weak experiments.

My main concern with this paper is the lack of novelty. One of the main known issues in relational learning (even out of tabular setting) is computational complexity and dealing with high dimensional data, however, the proposed method could not address this issue and only the experiments are conducted on tiny feature sizes which is not an issue. This is not consistent with real-world high-dimensional tabular data.

Apart from that, it is not clear to me why they have added the prior knowledge through regularization and not embedded it as a graph similar to MoReL. Incorporating as a regularizer could be done in simpler models as well and there is no comparison in the experiments.

**Summary Of The Review:**

I believe the paper is not novel and could not address the main issue of this domain. The experiments also cannot support the claims.

---

> ### Author Response · Authors · 2022-11-11
> **Response to Reviewer bxj6 (4/4)**
>
> ## Q5. Presentation of results
>
> *“Did not report the performance of raw data model…”*
>
> **Response**
> We described the evaluation procedure in Appendix B.1. Specifically, we reported utility as the relative difference in predictive performance, i.e. $AUROC_{synth} - AUROC_{test}$, where $AUROC_{synth}$ is the AUROC performance of a downstream model trained on synthetic data ($AUROC_{test}$ similarly defined). As you requested, we now report raw data performance in more detailed results tables in Appendix C.1 (see points and table below).
>
> *“Missed some of the basic regression methods…”*
>
> **Response**
> To evaluate utility, we reported average performance of three downstream models, which includes a linear regression model, as well as two-layer MLP, and XGB model (see Appendix B.1).
>
> *“reported an average of three different methods. I would like to see the performance of each separately”*
>
> *“AUROC is not sufficient… other metrics such as recall, FDR, F1, etc”*
>
> **Response**
> Thank you for these comments. We want to first re-affirm that our aim in reporting average performance is to provide a downstream task-, and model-agnostic evaluation to better evaluate the quality of the synthetic data.
>
> Since you are interested in the utility of individual downstream models, we provide a decomposition of the results below and in Appendix C.1, where we reported individual performance and included both AUROC and F1 results.
>
> [[**Updated**] Fine-grained utility results.](https://i.imgur.com/vTd5bRq.png)
>
> **Actions taken**
> We added utility results (as evaluated using both AUROC, and F1) for all baselines and raw data model in Table 6, Appendix C.1
>
> ---
> ## Q6. Reproducibility
>
> **Response
> As we stated in the reproducibility statement, we will release code publicly if this paper is accepted, to accompany the detailed implementation procedures in Appendix B.
>
> ---
> We thank the reviewer for your help in improving our work. Please let us know if our latest changes have addressed your concerns, and if there is anything else you would like to see.

---

> ### Author Response · Authors · 2022-11-11
> **Response to Reviewer bxj6 (3/4)**
>
> ## Q3. Learned structures
>
> *“Learned structure is not well explored”*
>
> **Response**
>
> As we have emphasized above, our goal is not structure discovery, but learning an approximate relational structure to generate better synthetic data (see Appendix D for more discussion). Structural discovery methods usually entail a very different set of assumptions, methods, and metrics to recover a relational structure.
>
> In our work, we have shown substantively that, even if the relational structure learned in GOGGLE is incorrect, generating wrt said structure can still improve generative performance! We highlight synthetic data quality of GOGGLE-COV and GOGGLE-BN in Table 3 (ablation study) and GOGGLE_50, and GOGGLE_100 in Figure 3 (prior knowledge). All four models learned structure that were approximately correct yet outperformed baseline models in synthetic data quality.
>
> Additionally, we want to remind the reviewer of visualizations and qualitative analysis of learned relational structure that already exists in Appendix C.2. Here, we further complement existing results with quantitative analysis, by including additional table reporting structural hamming distance [1] between the learned relational structure and the ground truth. The structural hamming distance (SHD) computes graph distances between predicted and ground truth graphs by the number of insertions, deletions or flips to transform one graph to another. We compare the relational structure learned by GOGGLE against 1) the graph learned by a Bayesian Network (**BN**), 2) Erdos-Renyi random graph generated with edge probability matching empirical edge probability p in the ground-truth graph (**ER(p)**), 3) thresholded correlation graph (**CORR**).
>
> [[Additional results] Quantitative analysis of learned structures.](https://i.imgur.com/btuHG1U.png)
>
> We describe results in the table above. We note that BN model is the *pseudo*-"oracle" as the true DGP for ECOLI and MAGIC are indeed Gaussian Bayesian Networks (matching the assumptions and model specifications of BN, giving BN the upper hand). ER(p) serves as a dummy baseline as the predicted graph is randomly guessed. We note that the correlation graphs (CORR, thresholded at 0.5) learn many spurious, yet uninformative edges as it is solely driven by association relationships. We note that the graphs learned by GOGGLE are more sparse, and informative.
>
> **Actions taken**
> We added the above table with structural distance metrics to complement analysis of learned relational structure in Appendix C.3.
>
> **References**
>
> [1] de Jongh, M. and Druzdzel, M.J., 2009. A comparison of structural distance measures for causal Bayesian network models. Recent Advances in Intelligent Information Systems, Challenging Problems of Science, Computer Science series, pp.443-456.
>
> ---
>
> ## Q4. Incorporating Prior Knowledge
>
> *“not clear… why prior knowledge through regularization and not embedded it as a graph”*
>
> **Response**
>
> As you stated, our approach (described in Section 5.2) is based on regularizing the learned graph $G$ with partial prior knowledge $G_0$, i.e. $\mathcal{R}_G=||G-G_0||_1$, this reflects the uncertainty that we have around variable dependencies (for ease of exposition, we refer to this as **setting A**). However, an alternative way to embed prior knowledge, if we are certain of it, is to use the prior graph directly, i.e. $M \odot G_0 + (1-M)\odot G$, where $M \in$ {$0$, $1$} is a mask of indicators for which variables we have certain prior knowledge on. Here, the entries in $M \odot G_0$ are not learned during training. We refer to this as **setting B**. We investigate the effect of including prior knowledge using this alternative mechanism in the table below. Evidently, setting B benefits from the higher certainty of having prior knowledge directly embedded into the relational structure, resulting in better synthetic data quality. Thank you for raising this comment.
>
> [[Additional results] Comparison of two forms of prior knowledge.](https://i.imgur.com/ee1WpqK.png)
>
> **Actions taken**
> We added to the descriptions in Section 5.2 to more accurately describe how prior knowledge is incorporated. Additionally, we included new results on alternative forms of prior knowledge in Appendix C.6

---

> ### Author Response · Authors · 2022-11-11
> **Response to Reviewer bxj6 (2/4)**
>
> ## Q2. Related works
>
> *“missed some related papers… including NRI, MoReL, BayReL and many other related relational inference papers”*
> *“self-supervised learning models… can be used as generative models, including VIME, SubTab”*
>
> **Response**
>
> Thank you for pointing us to the additional references. While we agree there are certain parallels to our work that should be highlighted, relational inference (incl NRI, MoReL, BayReL) and self-supervised methods for tabular data (incl. VIME, SubTab) are fundamentally different problems with different objectives, methods, and evaluation criteria. Therefore, we do not believe the related work takes away from the novelty of our proposed method (we highlight differences point-by-point in the table below).
>
> The goal of **relational learning** is to infer relationships between objects from observational data alone (interacting objects in NRI, and molecular interactions in multi-omics integration in BayRel and MoReL). The output of the learning method is a distribution over relational graphs, which is evaluated based on correctness of the recovered graph. The biggest difference to our work is that GOGGLE is fine if the structure is only **partially correct**, which is in stark contrast to any literature on structure discovery. The relational structure in our case acts as an inductive bias encouraging a sparse set of informative neighbors to be found, thus better learning the distribution of tabular data, and plays a regularization effect on spurious relationships. As we show in our response to Q3, these partially correct relational structures that we learn (with no guarantees of correctness) can consistently improve generative synthetic data performance.
>
> To the best of our knowledge, self-supervised VIME and SubTab **cannot** be used to generate synthetic data (although VIME can be used to generate augmented data by reconstructing a corrupted input). SSL methods for tabular data are similarly driven by uncovering relational information between features to learn good representations. However, this is often done implicitly, as in VIME, which applies a masking operator to encourage the representation learning module to learn inter-feature relationships. Similarly, SubTab employs a contrastive loss by generating views on subsets of features, implicitly encouraging learned representations to extract mutual information between features. Our work is different by incorporating a relational mechanism into the generative process, explicitly encouraging informative relations to be found, in addition to providing a flexible mechanism for prior knowledge and regularization. We discuss specific differences in objectives, methods and evaluation criteria in the table:
>
> [Comparison table.](https://i.imgur.com/ppQH7LQ.png)
>
> We hope our explanation has clarified the **distinction to existing work** and **novelty** of our method, although please let us know if you still have concerns.
>
> **Actions taken** We added suggested related works in Section 3, and in-depth discussions and table of comparison in Appendix D

---

> ### Author Response · Authors · 2022-11-11
> **Response to Reviewer bxj6 (1/4)**
>
> Thank you very much for your helpful feedback and comments. We aim to address all the individual points in your review here, but please also see the revised manuscript for changes (highlighted in blue).
>
> ---
> ## Q1. Novelty
>
> *“known issues in relational learning is computational complexity and dealing with high dimensional data… could not address this issue and experiments are conducted on tiny feature sizes… not consistent with real-world high-dimensional tabular data”*
>
> **Response**
>
> Thank you for this comment. We wish to re-iterate that the goal of this work is **not** relational inference. We aimed to make this abundantly clear in our response to Q2, where we clarify the exact differences in objectives, methods, and evaluation between our work in synthetic data and those in relational inference (and self-supervised learning). Then, in our response to Q3, we show that, while we are satisfied with learning an approximately correct relational structure, the resulting inductive bias consistently leads to better synthetic data quality.
>
> Indeed, recovering the exact relational structure does not guarantee the model can generate high quality synthetic data, with the latter being the focus of this work. We leverage relational learning as an inductive bias in our generative model to better learn the distribution of tabular datasets (akin to how CNNs/RNNs exploit relational information in homogenous data formats). As such, the key contributions of our work are in showing that 1) learning the relational structure leads to better modelling of the sparse, heterogeneous relationships common in tabular datasets, and that the relational structure is 2) a flexible mechanism to introduce regularization and 3) prior knowledge. We see these as three novel and important contributions to tabular generative models.
>
> We note that generating higher-dimensional synthetic data, while being a problem related to ours, involves a different set of considerations and tailored methods. We acknowledged this distinction in S6 Discussion: *“in this work, we focus on tabular data regime, where $d < 100$, future works can focus on higher-dimensional tabular data (e.g. genomics)... ”*. However, our method is still applicable to the vast majority of real-world tabular datasets, where $d < 100$ (for typical dimensionality of tabular datasets, see Table IV in [1], Table 1 in [2], and Table 1 in [3]). We evaluated our method on 10 datasets, ranging from 16-168 features, and 569 - 581,012 samples (see Table 4), which we consider representative of diverse datasets from different domains.
>
> **References**
>
> [1] Borisov, V., Leemann, T., Seßler, K., Haug, J., Pawelczyk, M. and Kasneci, G., 2021. Deep neural networks and tabular data: A survey. arXiv preprint arXiv:2110.01889.
>
> [2] Shwartz-Ziv, R. and Armon, A., 2022. Tabular data: Deep learning is not all you need. Information Fusion, 81, pp.84-90.
>
> [3] Gorishniy, Y., Rubachev, I., Khrulkov, V. and Babenko, A., 2021. Revisiting deep learning models for tabular data. Advances in Neural Information Processing Systems, 34, pp.18932-18943.

---

> > ### Comment · Reviewer_bxj6 · 2022-11-18
> > **I still believe the novelty and experiments are the main issues of the submission**
> >
> > Thank you for the responses. While the authors address some of my concerns, I still believe the novelty of this paper is limited and the authors could not address the main challenges in tabular data generation.
> >
> > The main issues of the tabular setting are 1) the heterogeneity and high dimensionality of the feature space as well as 2) the unstructured nature or unknown structure to be applied as a bias. Available methods on the generative domains that are based on VAE, GAN, etc., could easily handle the first problem, i.e. high dimensionality and heterogeneity of the feature space in their generation. However, they did not achieve a similar performance as the other domains like Image because there is no known structure to use as bias in their architecture.
> >
> > The proposed method tries to directly utilize relational learning (with a different objective that is not theoretically that much different) to solve the second issue, however, completely missed the broad range of tabular datasets and cannot even compete with the existing methods in this regard.
> >
> > In my view, the proposed method does not handle the problems but just reduces the search space of data/problem to a smaller set of datasets/problems. What I believe is interesting is in addition to the change of optimization objective of learning relations, making it scalable to the problems in your hand.
> >
> >
> > The authors' response also raised a new concern. The authors said: **while we are satisfied with learning an approximately correct relational structure, the resulting inductive bias consistently leads to better synthetic data quality.**
> >
> > If the authors observed that an approximately correct graph is enough, why not infer a graph from raw space with a simple pre-processing step and use it as a biased in the model? This is much simpler and applicable to high-dimensional data. I believe this should be included in the baselines to show introducing high computational complexity is reasonable. On a dataset like Adult, I believe you can have a reasonably good approximation of relationships throughout the raw data. On the other hand, when you have prior knowledge about the graph, what would be the results if one directly uses the available knowledge as a graph (not use it to regularize the relational learning)?
> >
> >
> > **I'm still not sure about the experiments:** As an example, the authors reported an average of three metrics (\alpha-precision, \beta-recall, authenticity) based on [1] as a metric for diversity comparison. Then, they also reported the fidelity of each model as a separate metric. As far as I can understand, 1) \alpha-precision is a fidelity metric that measures the rate by which the model synthesizes realistic-looking samples. What is the difference with the fidelity metric reported separately? And why here the authors reported an average of three metrics together? I would like to see the results for each metric separately. 2) \B-recall is just diversity and 3)authenticity is just generalization.
> >
> > Apart from that, I have asked the authors to report the results of each classifier separately. Thanks for the update. However, I still have questions regarding that. 1) In the revised version, the raw data works better in terms of the F1 score (see table 8). Would you please report the same F1 for all datasets not only two of them? Would you please elaborate on this observation? 2) It is not clear why the models are trained with D_test.
> >
> > In addition, the utility metric to me is very similar to **Machine learning efficacy** in CTGAN. Would you please describe the difference? If they are the same, why you defined a new metric, and if not, would you please include the metric which already has been used?

---

> > > ### Author Response · Authors · 2022-11-19
> > > **Response to New Comments (2/2)**
> > >
> > > ## Q4. Synthetic data metrics
> > >
> > > *“\alpha-precision is a fidelity metric that measures the rate by which the model synthesizes realistic-looking samples. What is the difference with the fidelity metric reported separately? And why here the authors reported an average of three metrics together?”*
> > >
> > > Thank you for this comment, we hope to clarify the distinction.
> > >
> > > $\alpha$-precision is a generalized fidelity metric proposed in [1], which assesses how well synthetic samples resemble real samples. The metric computes the probability that a synthetic sample resides in the $\alpha$-support of the real distribution, integrating over values of $\alpha \in [0, 1]$.
> > >
> > > The fidelity metric that is reported separately is *“AUROC performance of three post-hoc classifiers to distinguish real and generated samples”* (see **Section 5.1 Evaluation of Synthetic Data**). This is a *discriminative* score to discriminate between real and synthetic data at a **sample**-level as opposed to $\alpha$-precision, which is a statistical measure for fidelity at the **distribution**-level.
> > >
> > > We included both because the evaluation of synthetic data is an ongoing research problem, with no universally accepted metric, and including both provides a more comprehensive assessment of synthetic data quality. To make clear this distinction, we modify our metric names as follows: 1) **quality** (old name: diversity) now describes the average of $\alpha$-precision, $\beta$-recall, and authenticity; 2) **detection** (old name: fidelity) now describes the detection rate of real vs. synthetic samples.
> > >
> > > We averaged $\alpha$-precision, $\beta$-recall, and authenticity as a holistic score that represents overall quality of the synthetic data that can be easily interpreted through a single scalar value.
> > >
> > > **Actions taken**
> > > Revised manuscript with updated metric names.
> > >
> > > *"I would like to see the results for each metric separately."*
> > >
> > > Thank you for this comment. Since you are interested in the results for each metric individually, we provide a detailed decomposition of the quality metrics below and in Appendix C.1. We highlight that while many deep generative models achieve competitive $\alpha$-precision, they fail to capture the variability in the training data (*mode dropping*, see [1]) and does not generalize well (low authenticity). This is especially notable on Breast, where existing methods overfit to fewer samples, but our method better captures the variability and exploits sparsity to achieve better generalization performance.
> > >
> > > [Fine-grained quality metrics.](https://i.imgur.com/XHOIzyT.png)
> > >
> > > **References**
> > >
> > > [1] Sajjadi, M.S., Bachem, O., Lucic, M., Bousquet, O. and Gelly, S., 2018. Assessing generative models via precision and recall. Advances in neural information processing systems, 31.
> > >
> > > ---
> > >
> > > ## Q5. Downstream performance
> > >
> > > *“the utility metric to me is very similar to Machine learning efficacy in CTGAN. Would you please describe the difference?”*
> > >
> > > Utility evaluates the predictive performance of models trained on $D_{syn}$ and evaluated on $D_{test}$. We described this in **Appendix B.1 - Evaluation** *“To evaluate utility, we report average performance achieved by three downstream prediction models (linear model, two-layer MLP, and XGB model) trained on $D_{syn}$ and evaluated on $D_{test}$.”* *This is not a new metric*, but the same metric as *machine learning efficacy* reported in CTGAN, and as *classification/regression performance* reported in TableGAN.
> > >
> > > *“it is not clear why the models are trained with D_test”*
> > >
> > > We apologize for causing this confusion, there was a **typo** in the caption, which we have updated here and in the revised manuscript. We note that the original manuscript (see point above) correctly described the evaluation procedure.
> > >
> > > [[Updated] Fine grained utility results.](https://i.imgur.com/vTd5bRq.png)
> > >
> > > *“raw data works better in terms of the F1 score (see table 8). Would you please report the same F1 for all datasets not only two of them? Would you please elaborate on this observation?”*
> > >
> > > F1 is evaluated at a specific threshold, whereas AUROC is not. The difference between the two performance metrics on the Adult dataset is likely due to the imbalanced label distribution.
> > >
> > > We are happy to provide F1 scores for all datasets, but we hope you can understand that compiling and presenting results for all datasets as we have [done here](https://i.imgur.com/vTd5bRq.png) takes time, and will not be ready before the paper revision deadline, but **will** be ready for the camera-ready deadline. We will include the results in the camera-ready (if accepted).
> > >
> > >
> > > ---
> > >
> > > Thanks again for your help in improving our work! If our latest comments have addressed your concerns, we ask you to kindly consider increasing your score.

---

> > > ### Author Response · Authors · 2022-11-19
> > > **Response to New Comments (1/2)**
> > >
> > > Thanks a lot for the response, we really appreciate your engagement with our work, as we know this can be a very time consuming task! We collated your further comments into the following points and respond to them in turn.
> > >
> > >
> > > ---
> > >
> > > ## Q1. Dimensionality of the feature space
> > >
> > > *“main issues of the tabular setting are 1) the heterogeneity and high dimensionality of the feature space”*
> > >
> > > **Response**
> > >
> > > Thank you for bringing this point up again. While we definitely agree that generative modelling of high-dimensional tabular data is an important problem, *it is not the main focus or challenge* that we aim to address in this work. **Reviewer k5vp** and we agree that the paper’s scope–which concerns *“real-world tabular data (where typical dimensionality $d<100$)”*–is now clear in both abstract and intro. We wish to highlight that the datasets we used in our experiments are representative in terms of dimensionality for most machine learning practitioners in applications such as healthcare, finance, insurance, environmental, and sports science.
> > >
> > > We focused on a set of challenges that are more pertinent in this domain, which we see as equally important. In **Section 2 Challenges in Tabular Data Generation**, we laid out the three specific challenges that our method aims to address: 1) complicated relational structure underlying tabular data, 2) overfitting and memorization, and 3) incorporating domain knowledge. We addressed these challenges by introducing explicit relational inductive bias into the generative process.
> > >
> > > *“but just reduces the search space of data/problem to a smaller set of datasets/problems”*
> > >
> > > *“however, completely missed the broad range of tabular datasets and cannot even compete with the existing methods in this regard.”*
> > >
> > > The focus of our work is in modeling real-world tabular datasets that have dimensionality $d<100$. This focus is in line with existing works in tabular data generation (incl. CTGAN, TVAE, TableGAN). In fact, we evaluated our method on 3 of the 6 real-world datasets that are also used in CTGAN and TVAE (specifically Adult d=14, Covertype d=54, and Credit d=31). The other datasets employed in CTGAN, TVAE, and TableGAN are Intrusion d=41, News d=61, Health d=28, LACity d=21, Airline d=30. What we are trying to highlight is that the dimensionality we focus on is typical of real-world tabular datasets, and the 10 datasets we chose for evaluation, which have features ranging from $16$-$168$ (and sample sizes between $569$-$581012$) are appropriate and representative of the dimensionality encountered in these settings.
> > >
> > > Additionally, we highlighted in **Section 6 Discussions** that an important future step is to extend relational inductive bias to generative modelling in higher dimensions (e.g. genomics.) We hope that you can take into account the aforementioned challenges, motivations, and objectives when assessing novelty and contribution. Thank you again!
> > >
> > > ---
> > >
> > > ## Q2. Inferring graph as preprocessing
> > >
> > > *“why not infer a graph from raw space with a simple pre-processing step and use it as a biased in the model”*
> > >
> > > **Response**
> > >
> > > We agree. In fact, this is a key question that we addressed in our empirical investigations, i.e. the ablation study described in **Section 5.3 Ablation Study** (included in the original submission). Specifically, we considered three settings where a relational structure is first *‘infer from raw space’* as a *‘pre-processing step’*: 1) GOGGLE-ER (randomly initialized Erdos-Renyi random graph with 10% connectivity), 2) GOGGLE-COV (initialized using cross-correlation matrix), 3) GOGGLE-BN (initialized using PC algorithm). Essentially, we compared performance gained from learning the relational structure jointly with the functional relationships model, rather than learning the relational structure as a pre-processing step. We empirically observed in Table 4 that the joint learning of both components is crucial to achieve consistently good performance.
> > >
> > > ---
> > >
> > > ## Q3. Prior knowledge
> > >
> > > *“when you have prior knowledge about the graph, what would be the results if one directly uses the available knowledge as a graph”*
> > >
> > > **Response**
> > >
> > > Thanks for raising this suggestion again. In case you missed it, we would like to remind you of our **initial response to your comment in [Q4](https://openreview.net/forum?id=fPVRcJqspu&noteId=owyUK6z5JI)**. We evaluated this alternative form of incorporating prior knowledge and found that embedding available knowledge directly as a graph benefits from the higher certainty of prior knowledge (as compared to incorporating a regularizer), resulting in better synthetic data quality.
> > >
> > > We would also like to re-iterate that a wide variety of prior knowledge (e.g. graph types, connectivity) can be incorporated through regularization, which we already expanded on in **Appendix B.5 Prior Knowledge**.

---

> ### Author Response · Authors · 2022-11-16
> **Author Followup**
>
> Dear Reviewer bxj6,
>
> Once again, **thank you** for your thoughtful review. While we have this current period of discussion, please do let us know if our response has addressed your concerns - we are keen to keep engaging with you to address any additional questions or comments.
>
> Best wishes,
>
> The Authors

---

### Official Review · Reviewer_k5vp · 2022-10-26

**Confidence:** 3
**Correctness:** 3
**Technical Novelty And Significance:** 2
**Empirical Novelty And Significance:** 3
**Recommendation:** 8

**Clarity, Quality, Novelty And Reproducibility:**

The paper is overall clearly written and is easy to understand.
The quality of the proposed method seems reasonable; although it is not always the best, it most often is the best.
The proposed method for a model-based way to learn a generative model for tabular data is novel.
The work is NOT reproducible as code is not supplied (though there are some details in the appendix, I do not consider this reproducible unless code is provided).

A couple minor points to improve for paper clarify:
- I did not understand the point about synchronous vs asynchronous generation. There needs to be more explanation here, or it should go in the appendix (given that it seems the point was not revisited in the main text).
- How many samples are generated from Goggle for downstream evaluation for assessing Utility? Is it the same number of samples as there were original real samples? If so, it seems surprising to me that training on Goggle synthetic data is better than training on real data - this would be worth highlighting more!

**Strength And Weaknesses:**

Strengths:
- Empirically, the proposed method is usually the best performer across 10 benchmarks.
- The model-based approach to learning dependencies and graph-based approach to generative modeling is unique and a valuable contribution for the community. I could foresee future work building on this and finding better ways to incorporate prior information.
- Strong empirical evaluations, ablations, and benchmarking efforts to compare to baselines overall.

Weaknesses:
- The story on leveraging prior information is not convincingly demonstrated through experiments. Looking at Figure 3, Goggle never seems to outperform BN, and in some cases is handily beat by either BN or CTGAN. Given that leveraging priors is a major strength of the proposed method, the empirical results here are not consistent with that story. Also, the language in that section "it is encouraging to see that our method achieves comparable performance to BN methods" is misleading and should be altered.
- Placement of results tables. Though the method is evaluated on 10 datasets, only results on 4 datasets are presented in the main text. From an outside perspective, it would appear as if these datasets may have been cherrypicked, especially since relative performance numbers on the other 6 datasets in the appendix are not as high.
- Baselines comparison. Was equal hyperparameter tuning and equal computational budget given to each of the baselines? It does not seem specified in the appendix.
- Comparison to a MM (mixture models) baseline is missing.

**Summary Of The Paper:**

This work introduces a new generative model for tabular data. It proposes to learn both a relational structure between variables and also a data generating process conditioned on that structure. This formulation has not been presented before for neural methods and is a significant contribution. The benefits of this method are that it can incorporate prior information via an adjacency matrix, leverage the benefits of neural networks for generation in the large data regime, and is more amenable to certain types of regularization. These benefits are quantified empirically on a few tabular datasets where the proposed method seems to be the most consistent performer and usually outperforms other baselines.

**Summary Of The Review:**

Overall, the proposed method in this paper seems very promising - a model-based tabular generation scheme that can leverage prior knowledge, that is more amenable to regularization in the small data regime, and that has solid performance empirically across a number of datasets. However, the paper does not meet two of these major marks:
1) In terms of leveraging prior information, it seems Goggle was always outperformed by either BN or CTGAN in this setting.
2) Details of hyperparameter tuning are not provided. Specifically, it seems like an equal computational budget may not have been given to each method. Thus, it is hard as a reviewer to deduce whether performance gains come from the superiority of the method or from more extensive parameter tuning.
Nonetheless, the effort put forth in the paper is solid and the main ideas presented would make a valuable contribution. I am marking my score a little lower due to the two major drawbacks above, but I am happy to increase the score during & after discussion with the authors, depending on their response.

********* UPDATE AFTER REBUTTAL ****************

The authors have resolved my main concerns regarding hyperparameter tuning and have adjusted the language in the prior information section to scope down the claims (as well as adding some new experiments). Additionally, the authors have added a clarification that their method has only been validated on relatively low-dimensional datasets, which their experiments support. For this reason, i am increasing my score as I believe this is a solid contribution to the community.

---

> ### Author Response · Authors · 2022-11-11
> **Response to Reviewer k5vp (2/2)**
>
> ## Q4. Mixture model baseline
> *“comparison to MM baseline is missing”*
>
> **Response**
>
> Thank you for pointing this out, we added results on synthetic data quality generated using mixture model baseline in Table 2, and include implementation details in Appendix B.2. We find that GOGGLE significantly outperforms MM baseline on all metrics, as the parametric MM is limited in modelling more complex dependencies.
>
> ---
> ## Q5. Asynchronous generation
> *“the point about synchronous vs asynchronous generation”*
>
> **Response**
>
> Here, synchronous vs asynchronous describes the message passing (MP) scheme and how it relates to the generative process, a synchronous MP scheme sees messages exchanged between all nodes in the graph (variables) simultaneously so all variables are generated jointly. In contrast, messages are exchanged sequentially in asynchronous MP, so variables are generated sequentially. In our synthetic data model, we employ a synchronous scheme to jointly generate all features in a synthetic sample, whereas in our data augmentation model, which learns an underlying DAG, messages are generated sequentially, such that each variable is generated conditionally only on its parents (see Appendix A).
>
> **Actions taken**
> We added a more detailed explanation about synchronous vs. asynchronous generation in Appendix A.
>
> ---
> ## Q6. Number of samples
>
> *"how many samples are generated from GOGGLE for downstream evaluation for assessing utility?”*
>
> **Response**
> Thanks for this question! As we described in Appendix B.1 - Evaluation, we generate a synthetic dataset $D_{syn}$ that has the same number of samples as $D_{train}$ for a fair comparison on evaluating utility.
>
> ---
> ## Q7. Reproducibility
>
> **Response**
> As we stated in the reproducibility statement, we will release our code to the public if this paper is accepted.
>
> ---
> We thank the reviewer for your help in improving our work. Please let us know if our latest changes have addressed your concerns, and if there is anything else you would like to see.

---

> ### Author Response · Authors · 2022-11-11
> **Response to Reviewer k5vp (1/2)**
>
> Thank you very much for your helpful feedback and comments. We aim to address all the individual points in your review here, but please also see the revised manuscript for changes (highlighted in blue).
>
> ---
>
> ## Q1. Prior information
> *“figure 3, GOGGLE never seems to outperform BN, and in some cases is handily beat by either BN or CTGAN”*
>
> **Response**
>
> Thank you for this comment. We hope to clarify that the quality of synthetic data (shown in Figure 3) is evaluated in terms of **diversity** (higher is better), **fidelity** (lower is better), and **utility** (higher is better) [1]. As Figure 3 depicts, GOGGLE_50 (partial knowledge) and GOGGLE_100 (full knowledge) outperforms neural baselines (CTGAN, NF) on both ECOLI and MAGIC and across all four data sizes considered. We have redrawn the figure to avoid misleading the viewer.
>
> [Redrawn Figure 3.](https://i.imgur.com/CbVcF82.png)
>
> The DGPs for both datasets (known a-priori) are Gaussian Bayesian Networks (i.e. edge relations are linear Gaussian). This matches exactly the assumptions of the BN baseline (which fits parametric Gaussian edges), giving BNs the upper hand as they have the correct model specification. To further understand the impact of prior knowledge on synthetic data quality in more realistic settings, we evaluate on a simulation, where the underlying graph is a BN (same graphs employed in ECOLI) but we replace the Gaussian functional relations with highly non-linear relationships, parameterised by a single hidden-layer MLP. For clarify, we refer to this semi-synthetic dataset as modifiedECOLI.
>
> We believe this setting should play more to the expressiveness of our decoder. Indeed, we find that introducing GOGGLE with prior knowledge achieves superior diversity and fidelity on modifiedECOLI.
>
> [[Additional results] Prior knowledge and synthetic data quality on modifiedECOLI.](https://i.imgur.com/LIO1dVI.png)
>
> We hope that you find the new Figure 3 easier to parse, and the new set of results more compelling, although please let us know if you still have concerns.
>
> **Actions taken**
> We redrew figure 3 for better presentation of our prior knowledge results. Furthermore, we provide additional results on semi-synthetic modifiedECOLI in Table 12, Appendix C.6.
>
> **References**
>
> [1] Alaa, A., Van Breugel, B., Saveliev, E.S. and van der Schaar, M., 2022, June. How faithful is your synthetic data? sample-level metrics for evaluating and auditing generative models. In International Conference on Machine Learning (pp. 290-306). PMLR.
>
> ---
> ## Q2. Results presentation
> *"only results on 4 datasets are presented in the main text… these datasets may have been cherrypicked”*
>
> **Response**
>
> We want to stress that it was not our intention to cherry pick favorable results. We presented a subset of results in the main text due to space considerations! To give a more holistic summary of performance, we add a set of summary statistics (see below, and in Table 6 of revised manuscript) that report the average rank of each method across all datasets evaluated.
>
> [Average rank of considered methods (synthetic data quality).](https://i.imgur.com/FzOo5nf.png)
>
> We highlight that GOGGLE achieves the best quality **9/10** times, the best fidelity **6/10** times, and the best utility **7/10** times, often with no outstanding runner-up. The summary statistics demonstrate that GOGGLE consistently outperforms other baselines.
>
> **Actions taken**
> We reported average rank of each method on all datasets in Table 6, Appendix C.1
>
> ---
> ## Q3. Baseline comparison
>
> “was equal hyperparameter tuning and equal computational budget given to each of the baselines”
>
> **Response**
>
> Yes, during hyperparameter search, each baseline is allowed 10 iterations using Bayesian Optimization [1], where the search objective is the reconstruction loss on the validation set. We believe that this procedure presents a fair comparison (see [1, 2]). We add the exact hyperparameters considered for each baseline in Appendix B.
>
> **Actions taken**
> We added hyperparameter tuning details in Appendix B.
>
> **References**
>
> [1] Shahriari, B., Swersky, K., Wang, Z., Adams, R.P. and De Freitas, N., 2015. Taking the human out of the loop: A review of Bayesian optimization. Proceedings of the IEEE, 104(1), pp.148-175.
>
> [2] Snoek, J., Larochelle, H. and Adams, R.P., 2012. Practical bayesian optimization of machine learning algorithms. Advances in neural information processing systems, 25.

---

> > ### Comment · Reviewer_k5vp · 2022-11-16
> > **Prepared to increase score given acknowledgements**
> >
> > Thank you for the detailed explanations and additional results.
> >
> > The response has alleviated my concerns with respect to results presentation, baselines comparisons, MM baseline, asynchronous generation, number of samples, and reproducibility.
> >
> > In terms of Q1 prior information, thanks for the updated figure 3. To me, it still seems like BN handily beats Goggle on diversity, while fidelity and utility is a tossup between the two. The new results on modifiedECOLI are appreciated do showcase Goggle's strength better. I think the text needs to be more reflective of the results - "it is encouraging to see that our method achieves comparable performance to BN methods" needs to be a lot more nuanced.
> >
> > After reading the other reviews, Reviewer bxj6 has brought up an important point I missed that all tabular datasets used are low-dimensional. I think this is an important point and needs to be acknowledged early (but the fix here is simply scoping/adjusting claims).
> >
> > I am prepared to increase my score, conditional on 3 changes being made for camera ready:
> > 1. Acknowledge in the abstract AND intro that the proposed method is validated on *low-dimensional* tabular datasets.
> > 2. Adding Table 6 to the main text (or presenting the average rank of each method in some way in the main text).
> > 3. Scoping down the language in Section 5.2 to more accurately reflect nuance in the results.
> >
> > Authors, please acknowledge this if you agree, or follow up with more discussion if you do not.

---

> > > ### Author Response · Authors · 2022-11-16
> > > **Thank you**
> > >
> > > Dear Reviewer k5vp,
> > >
> > > Thanks a lot for the response, we really appreciate your engagement with our work, as we know this can be a very time consuming task!
> > >
> > > We agree with your 3 recommended changes, and have highlighted in <span style="color:teal">teal</span> our changes to the manuscript. These changes **will be incorporated** in the camera ready. Specifically:
> > >
> > > * Clarified that we focus on tabular data with typical dimensionality ($d<100$) in the abstract AND introduction,
> > > * Inserted Table 3 into Section 5.1 of the revised manuscript to report average method rank,
> > > * Adjusted writing in Section 5.2 to better reflect results.
> > >
> > > Thank you again for agreeing to increase your score.
> > >
> > > The Authors

---

> > > > ### Comment · Reviewer_k5vp · 2022-11-17
> > > > **score increased**
> > > >
> > > > please see the updated review for my reasoning

---

> > > > > ### Author Response · Authors · 2022-11-18
> > > > > **Thank you again**
> > > > >
> > > > > Dear Reviewer k5vp,
> > > > >
> > > > > We are glad to have addressed your concerns, thank you for your help in improving our work.
> > > > >
> > > > > Best regards,
> > > > >
> > > > > The Authors

---

### Public Comment · ~Tianping_Zhang1 · 2023-02-17
**Are the codes and datasets available?**

Thank you for the interesting work! I was wondering if the codes for both the GOGGLE method and the baseline methods are available anywhere. I couldn't seem to find them in the supplementary materials. It's important for reproducibility purposes, so I'd greatly appreciate it if you could provide me with the codes.

---

> ### Author Response · Authors · 2023-03-10
> **Yes!**
>
> Dear Tianping,
>
> Thanks for engaging with our work! The code repository is linked in the manuscript, and can be found here: https://github.com/tennisonliu/GOGGLE, as well as the lab repository: https://github.com/vanderschaarlab/GOGGLE.
>
> Best,
>
> The Authors

---

### Decision · Program_Chairs · 2023-01-20

**Decision:**

Accept: poster

**Justification For Why Not Higher Score:**

A broader set of datasets (e.g. higher dimensional) might help to make this paper of interest to a wider subset of the community.

**Justification For Why Not Lower Score:**

The experimental comparison of an MLP decoder vs a GNN encoder makes a potentially useful contribution.

**Metareview: Summary, Strengths And Weaknesses:**

This paper adds to the emerging literature on deep networks for tabular data. This paper focuses on generative models, with the goal being specifically to generate synthetic data that has similar properties to a real tabular dataset. The paper proposes a VAE-based approach, in which the decoder features a graph neural network, with a node for every variable.

There was significant disagreement regarding the paper. The AC and the reviewers engaged in an active discussion, including an in-person meeting, where we discussed the points from the contrasting reviews and the author response. This meeting clarified the issues, but did not yield consensus. For this reason, I carefully reviewed the paper, reviews, and author response in order to make my recommendation.

The reviewers agreed that the architecture is conceptually interesting. The reviewers praised the breadth of the experimental evaluation, particularly in the choice of baselines.

One reviewer cited the “limited novelty” --- I would agree with this, if it is taken to refer to the statement that essentially what the paper is doing is modifying the decoder of a VAE from an MLP to a GNN --- this does not seem a large change, so one could argue that this paper is technically fairly similar to existing work. However, a priori, it would not be clear to me that a GNN would be beneficial for this task, maybe a MLP is good enough. Therefore, I believe this experimental demonstration makes a valuable contribution, especially when (as here) the graph structure is learned as well.

About the evaluation metrics: I agree with reviewer bxj6 that it does not seem meaningful to average alpha-precision, beta-recall, and authenticity from Alaa et al. Nor does it seem right to present these as orthogonal dimensions to the detection AUC. This can be seen from the authors’ definition of quality vs detection (“samples should be realistic” vs “samples should be indistinguishable from the real data”). How are these concepts different? Surely if a sample is realistic, it is difficult to distinguish from the real data, and vice versa? I agree with the reviewer that it would be better to present the three quality methods separately. I strongly encourage the authors to do so. I think the other two metrics in this paper are reasonable, even if I am not sure if the detection metric provides additional information beyond that in alpha-precision and beta-recall.

Overall though, my opinion is that this shortcoming of the paper is not serious enough to seriously jeopardize the main findings of the paper, so I recommend that this paper be accepted.


Minor comments:

Thanks for adding the mixture model (MM) baseline. I believe that you need to add it to the list of “benchmarks” in Section 5. Also, to be fair, there should be an option for the number of mixture components that causes the number of parameters (i.e., means, variances, mixture weights) to roughly match those of the neural networks. Has that been done?


**Note From Pc:**

if the above contains the word "oral" or "spotlight" please see: "oral" presentation means -> notable-top-5% and "spotlight" means -> notable-top-25%. As stated in our emails, we are disassociating presentation type from AC recommendations

**Summary Of Ac-Reviewer Meeting:**

The metareview enumerates the points that were discussed at the AC-reviewer meeting.